# Observations and modelling of algal growth on a snowpack in northwest Greenland

Yukihiko Onuma[1], Nozomu Takeuchi[2], Sota Tanaka[2], Naoko Nagatsuka[3], Masashi Niwano[4], Teruo Aoki[5,4]

[1] Institute of Industrial Science, The University of Tokyo, Kashiwa, 277-8574, Japan
[2] Graduate School of Science, Chiba University, Chiba, 263-8522, Japan
[3] National Institute of Polar Research, Tokyo, 190-8518, Japan
[4] Meteorological Research Institute, Japan Meteorological Agency, Tsukuba, 305-0052, Japan
[5] Graduate School of Natural Science and Technology, Okayama University, Okayama, 700-8530, Japan

*Correspondence to*: Yukihiko Onuma (onuma@iis.u-tokyo.ac.jp)

**Abstract.** Snow algal bloom is a common phenomenon on melting snowpacks in polar and alpine regions and can substantially increase melting rates of the snow due to the effect of albedo reduction on the snow surface. In order to reproduce algal growth on the snow surface using a numerical model, temporal changes in snow algal abundance were investigated on the Qaanaaq Glacier in northwest Greenland from June to August 2014. Snow algae first appeared at the study sites in late June, which was approximately 94 hours after air temperatures exceeded the melting point. Algal abundance increased exponentially after the appearance, but the increasing rate became slow after late July, and finally reached $3.5 \times 10^7$ cells m$^{-2}$ in early August. We applied a logistic model to the algal growth curve and found that the algae could be reproduced with an initial cell concentration of $6.9 \times 10^2$ cells m$^{-2}$, a growth rate of 0.42 d$^{-1}$, and a carrying capacity of $3.5 \times 10^7$ cells m$^{-2}$ on this glacier. This model has the potential to simulate algal blooms from meteorological data sets and to evaluate their impact on the melting of seasonal snowpacks and glaciers.

## 1 Introduction

Snow algae are cold-tolerant, photosynthetic microbes growing on snow and ice and are commonly found on glaciers and snowfields worldwide. Snow algal blooms occur on thawing snow surfaces and change the colour of the snow to red or green (Thomas and Duval, 1995; Hoham and Duval, 2001; Takeuchi, 2013). Red snow algal blooms (usually by *Chlamydomonas* (*Cd.*) *nivalis*) commonly occur in polar and alpine snow fields (Hoham and Duval, 2001; Segawa et al., 2005; Takeuchi et al., 2006; Lutz et al., 2016; Tanaka et al., 2016; Ganey et al., 2017).

The conditions required for the growth of snow algae are the occurrence of liquid-water, solar radiation, and nutrients. Snow algal cells are typically present in the liquid water film surrounding snow grains when the snow melts (Fukushima, 1963). Field observations showed that snow algae begin to grow when the air temperature is above the freezing point for several days, suggesting that algal growth requires a certain amount of water content in the snow (Pollock, 1972; Onuma et al., 2016). A field study on algal photosynthesis suggested that algal growth requires at least 1% of incident photosynthetically active

radiation in the snowpack, promoting photosynthesis and germination of algae (Curl et al., 1972). After the algae appears on the snow surface, nutrient depletion (particularly nitrates) in the snowpack can cause shifts in life cycle phases and decrease the growth rate (Hoham et al., 1989). Previous studies have shown that the abundance of snow algae increases as the snow melts. For example, snow algal abundance on a glacier in Alaska continued to increase during the melting season until the

snowpack completely melted on the glacier surface (Takeuchi, 2013). Snow algal abundance on a seasonal snowpack in Japan increased exponentially with snow melting until the snowpack completely melted (Onuma et al., 2016). Such temporal changes in snow algal abundance can be affected by the snow conditions, such as water content, solar radiation, and nutrient availability.

    A numerical model could be utilized to reproduce the seasonal change in algal abundance on snowpacks, to understand algal growth in snowfields on a regional or worldwide scale, and to evaluate their effects on the surface albedo and resultant melting

rate. The effect of algae on surface albedo can be physically calculated using an albedo model based on algal abundance (Cook et al., 2017a; 2017b). A temporal change in snow algal abundance could also be reproduced using a numerical model. Many models have been proposed and applied to temporal changes in the abundance of photosynthetic microbes in aquatic environments such as lakes or oceans. For example, there has been a model for cyanobacteria in lakes, which can reproduce their exponential growth using their initial concentration, growth rate, and nutrient concentration (Chen et al., 2009).

Additionally, a model for algae (diatoms) growth in sea ice was developed using a sea ice physical model (Pogson et al., 2011). This model can reproduce the temporal change in chlorophyll a concentration in Arctic sea ice from the initial chlorophyll a concentration, algal growth rate, and grazing rate. The exponential growth of snow algae observed on a seasonal snowpack in Japan was reproduced using a Malthusian model (Onuma et al., 2016). Although this model might be effective for the seasonal snowpacks that exist for a short period and disappear in spring or early summer, it is questionable whether the model is suitable

for algae on permanent snowfields or glaciers.

    The Greenland Ice Sheet, the second largest continuous body of ice in the world, is known to be inhabited by snow algae. Several studies have reported the visible red snow caused by blooms of *Cd. nivalis* over the ice sheet (Lutz et al., 2014; Uetake et al., 2010; Takeuchi et al., 2014). The ice sheet is reportedly losing mass due to an increase in temperature and decrease in surface albedo during the last two decades (Rignot et al., 2008; Wientjes and Oerlemans, 2010; Box et al., 2012). Decline in

surface albedo by snow and ice algal blooms can increase surface melting rates and thus is likely one of the factors to cause mass loss of the ice sheet in recent years (Yallop et al., 2012; Aoki et al., 2013; Lutz et al., 2014; Lutz et al., 2016; Tedstone et al., 2017; Stibal et al., 2017). Observation of a glacier in south east Greenland showed surface reflectance in the visible wavelengths for red snow (49%) to be lower than that of clean snow (75%), and that snow algal growth might lead to a positive feedback, increasing the melting rate of the glacier (Lutz et al., 2014). Quantification of snow algal abundance is important for

estimating the melting rate of snow over the ice sheet. Niwano et al. (2015) demonstrated that the snow albedo and snow melting in Greenland Ice Sheet can be simulated by a snow physical model (Niwano et al., 2012) that incorporates a physically based snow albedo model (Aoki et al., 2011). Establishment of a numerical model for algal growth possibly lead to simulate the snow melting including the effect of algal growth on snow albedo by coupled snow microbial-physical model. However,

there is little information on the temporal changes in snow algal abundance on a snowpack in Greenland, and a numerical model for the snow algal growth has not been established to date.

In this study, biological and meteorological observations were conducted on the Qaanaaq Glacier located in north west Greenland in order to quantify the temporal change in snow algal abundance and establish a numerical model for algal growth. Temporal changes in algal abundance on the snow surface were quantified at two locations on the glacier from June to August in 2014 and were fitted to a simple numerical equation. Factors affecting the parameters of the equation are discussed in terms of meteorological data and, physical and chemical snow data from the study sites.

## 2 Study sites and methods

The investigation was conducted at the Qaanaaq Ice Cap in northwest Greenland (Fig. 1) from June to August in 2014. The Qaanaaq Ice Cap, which lies on a small peninsula of north west Greenland, covers an area of 286 km$^2$ and has an elevation of approximately 1,110 m a.s.l. (Takeuchi et al, 2014; Sugiyama et al, 2014). We selected two study sites at different elevations (Sites-A and B) on the Qaanaaq Glacier, which is an outlet glacier of the ice cap and is easily accessible on foot from Qaanaaq village. Site-A is a snowpack located at an elevation of 551 m a.s.l. towards the middle of the glacier and is likely formed by snowdrift. Since the depth of the snowpack was deeper than that of surrounding areas, the snow persisted through much of the melting season. Site-B is located at 944 m a.s.l., and was close to the equilibrium line of the glacier (Tsutaki et al., 2017). Meteorological data used in the study were collected with an automatic weather station (AWS), which was installed at Site-B in 2012 by the Snow Impurity and Glacier Microbe effects on abrupt warming in the Arctic project (SIGMA) (Aoki et al., 2014). Air temperature and solar radiation were collected hourly from April to August 2014 using the AWS. Aoki et al. (2014) provided a more detailed description of the AWS. The temperature sensor and pyranometer of the AWS were placed at heights of 3.0 m and 2.5 m above the snow surface, respectively. Air temperature at Site-A was calculated from the air temperature collected at Site-B with a temperature lapse rate, which was assumed as $-7.80 \times 10^{-3}$ K m$^{-1}$ (Sugiyama et al., 2014). Solar radiation at Site-A was measured hourly from day 172 (21 June 2014) to 214 (2 August 2014) with a pyranometer (EKO ML-020) installed at 1.5 m above the snow surface. The measured time of the hourly meteorological data is defined as Local Time (LT = Greenwich Time $-$ 2 h) in summer.

Snow pits were observed once weekly during the study period at both sites to determine vertical profiles of snow type, temperature, density, and liquid-water content. The snow temperature was measured with a thermistor sensor (CT-430WP, Custom Ltd, Tokyo, Japan). The volumetric liquid-water content in snow layers was obtained from snow density and snow permittivity, which were measured using a density sampler and dielectric probe (Denoth, 1994), respectively. Snow surface temperature was obtained from direct measurements and from calculating the observed downward and upward longwave radiant fluxes assuming the emissivity of the snow surface to be 0.98 (Armstrong and Brun, 2008), following the protocol of Niwano et al. (2015).

Surface snow collection and snow pit observation were performed, simultaneously from days 162 to 214 (nine times total) at Site-A and from days 168 to 215 (eight times total) at Site-B. Samples were collected from one to five randomly selected surfaces (depth of 0 cm to 2 cm) using a stainless-steel scoop. The sampling area ranged from 100 to 900 cm$^2$ and was recorded for each collection. Snow layers below the surface were also collected from snow pits at Site-A on day 162 and Site-B on day 168. The samples collected were from the surface layer (depth = 0 – 2 cm), the subsurface layer (depth = 2 – 10cm), and the layers of every 10 cm down to the previous summer layer (depth = 150 cm for the site-A and 142 cm for the Site-B). All of the samples were preserved in Whirl-Pak® bags (Nasco, Fort Atkinson, Wisconsin, USA). Electrical conductivity (EC) and pH for the collected samples were measured using a portable pH-conductivity meter (F-54, HORIBA, Japan) after the samples were melted in Qaanaaq village. Samples used for algal cell analysis were collected separately. These samples were melted and preserved in 3% formalin in 30 ml clean polyethylene bottles before being transported to Chiba University, Japan, for analysis.

Algal abundance was obtained by cell counting and was represented as cell numbers per unit surface area of snowpacks. Water samples of 20–1000 µl were filtered through a hydrophilized PTFE membrane filter (pore size 0.45 µm, Millipore). The number of algal cells on the filter was counted two to five times for each sample using an optical microscope (BX51, OLYMPUS, Japan) and cell concentration (cells L$^{-1}$) were obtained from mean cell counts and filtered sample volumes. Cell numbers per unit area (cells m$^{-2}$) were calculated using the cell concentration and area of sample collection. To obtain a cell volume biomass (biovolume), mean cell volumes were estimated by measuring the size of 5–50 cells for each species using a microscope and mean cell volume was obtained geometrically. Total algal volume per unit area (mL m$^{-2}$) per taxon was obtained by the multiplying cell count and cell volume.

Abundance of mineral particles in snow was quantified using another set of samples collected from the snow surface. Melted samples were dried (60°C, 24 h) in pre-weighed crucibles then combusted (500°C, 3h) in an electric furnace to remove organic matter. The mass of mineral particles per area (g m$^{-2}$) was obtained from the combusted sample weight and sampling area since only mineral particles remained after combustion.

## 3 Results

### 3.1 Meteorological conditions

Meteorological observations on the Qaanaaq Glacier showed that air temperature was below 0°C from April through most of June and increased above 0°C from late June through early August (Fig. 2a). The daily mean air temperature at Site-B ranged from -25.8°C to -11.1°C in April and from -19.7°C to -7.9°C in May. It first exceeded 0°C in daytime on day 154 (3 June 2014) and remained above 0°C from late June to early August. This air temperature record indicates that snow melting occurred continuously from late June to August at the study sites.

Solar radiation gradually increased from April to mid-July, before decreasing (Fig. 2b). At the location of the Qaanaaq Glacier, the sun never set from day 108 (18 April 2014) to day 241 (29 August 2014). Monthly mean solar radiation at Site-B

for April, May, June, and July was 165, 276, 296, and 244 W m$^{-2}$, respectively. The daily mean solar radiation in July ranged from 71 and to 375 W m$^{-2}$ (mean: 218 W m$^{-2}$) and from 90 to 383 W m$^{-2}$ (mean: 244 W m$^{-2}$) at Sites-A and B respectively, indicating that the solar radiation did not significantly vary among sites.

### 3.2 Physical and chemical conditions of surface snow

Snow observations showed that the surface snow was consistently wet from late June to early August at both sites (Figs. 3 and 4). When the observation was started on day 168 (17 June 2014), the surface snow at Site-B was fresh dry snow without surface melt. This snow became granular on day 176, implying that the snow surface began to melt. Surface snow density was 386 kg m$^{-3}$ on day 168 and gradually increased until day 215 (3 August 2014, 489 kg m$^{-3}$). The mean snow grain size was 0.3 mm on day 168, 0.9 mm on day 181, and varied between 0.7 and 0.9 mm until day 215. Snow surface level decreased by 121 cm

during the study period (47 days). Surface snow temperature was –0.2°C on day 168 and 0°C from days 176 to 215, except day 197 (–0.1 °C). Hourly surface snow temperature was calculated from longwave radiation and showed that the duration temperature was above 0°C for 885 h of 1129 h during the study period. The volumetric liquid-water content of surface snow was 6.3% on day 181 and varied between 3.8 and 4.9% until day 215. Changes in snow properties were similar between sites. For example, the surface snow of Site-A was granular and the surface snow temperature was 0°C after day 179. The results

indicate that the snowpacks at both sites melted continuously from late June until early August.

The mass of mineral particles in surface snow gradually increased from June to early August at both sites, and it was consistently greater at the lower site (Site-A) than at the higher site (Site-B) (Figs. 3 and 4). The mineral abundance at Site-B was $3.0 \times 10^{-3}$ g m$^{-2}$ on day 176 and gradually increased to $7.6 \pm 3.0 \times 10^{-1}$ g m$^{-2}$ (mean ± SD) until day 215. The abundance at Site-A was 1.4 g m$^{-2}$ on day 179 and gradually increased to $6.6 \pm 1.9$ g m$^{-2}$ until day 214. A statistical test demonstrated that

the temporal changes in mineral abundance were significant at both sites (one-way ANOVA, Site-B: F = 4.95, P = 0.02 < 0.05; Site-A: F = 2.74, P = 0.004 < 0.01). The comparison of the mineral abundance in August between Sites-A and B showed that their difference was statistically significant. ($6.6 \pm 1.9$ vs. $7.6 \pm 3.0 \times 10^{-1}$ g m$^{-2}$; Student's $t$-test, $t$ = 4.10, P = 0.009 < 0.01).

The EC and pH of surface snow did not show seasonal trends or differences between the sites. The EC ranged from 1.5 to 4.0 µS cm$^{-1}$ (mean: 2.7 µS cm$^{-1}$) and from 0.4 to 3.2 µS cm$^{-1}$ (mean: 2.4 µS cm$^{-1}$) at Sites-A and B, respectively. The pH ranged

from 5.5 to 6.2 (mean: 5.9) and from 5.3 to 6.1 (mean: 5.8) at Sites-A and B, respectively. There was no significant difference in EC or pH between sites in late June (Student's $t$-test, EC: $t$ = 2.47, P = 0.13 > 0.05; pH: $t$ = 2.32, P = 0.15 > 0.05). The EC and pH in July and August did not significantly vary among sites.

### 3.3 Snow algae on snow surface

Microscopic observation revealed that the red spherical algal cells were dominant at both study sites. Algal cells (Fig. 5)

contained a reddish-orange and/or green pigment and were $21.3 \pm 2.3$ µm in diameter. The cell volume biomass of this alga accounted for over 95% of the total algal biomass at both study sites. This alga was likely *Chlamydomonas* (*Cd.*) *nivalis* since

the shape, size, and pigmentation (Fig. 5) corresponded with the taxon observed previously in 2007 and 2012 on this glacier (Uetake et al., 2010; Takeuchi et al., 2014).

The other cell types were present in the samples in trace amount. One was spherical in shape with orange or green pigment, and its cell size was smaller ($9.0 \pm 2.2$ µm) than the previously described algae. Another cell was also spherical but with pale blue-green pigments and was much smaller in size ($4.6 \pm 1.2$ µm). These types of algal cells were likely the undefined alga and *Chroococcaceae cyanobacterium* reported by Uetake et al. (2010), respectively.

### 3.4 Temporal changes in algal cell concentration of surface snow

Microscopic analysis revealed that the algal cell appeared on the surface snow in late June and gradually increased until late July (Fig. 6). Algal abundance was 7.4 cells $m^{-2}$ at Site-B when the algae first appeared on day 181 and then increased to $5.0 \times 10^5$ cells $m^{-2}$ until day 215, although abundance decreased occasionally on days 190 and 197 (Fig. 6b). The algal cells at Site-A first appeared on day 179 ($3.1 \times 10^3$ cells $m^{-2}$), then their abundance exponentially increased until day 201 ($2.2 \times 10^7$ cells $m^{-2}$) (Fig. 6a). Temporal changes in algal abundance were significant at both sites (one-way ANOVA, Site-A: F = 2.45, P = 0.006 < 0.01; Site-B: F = 2.91, P = $5.9 \times 10^{-5}$ < 0.01). The snow pit samples collected before the appearance of the algae (on days 162 at Site-A and 168 at Site-B) contained no algal cell at in all of the snow layers down to the last summer surface.

The algal abundances on the snow surface at Site-B continued increasing until early August, whereas the abundances at Site-A did not significantly increase between days 201 to 214 (Fig. 6). The mean algal abundance at Site-A was $2.2 \times 10^7$ cells $m^{-2}$ on day 201 and $3.5 \times 10^7$ cells $m^{-2}$ on day 214. The algal abundance on day 201 was 637 times that of day 195; however, algal abundance on day 214 was only 1.6 times that of day 201. The temporal change in algal abundance at Site-A was not significant between days 201 and 214 (one-way ANOVA, F = 4.56, P = 0.26 > 0.01).

## 4 Discussions

### 4.1 Origin of snow algae and their growth condition on the Qaanaaq Glacier

The red snow phenomenon observed on the Qaanaaq Glacier is likely to occur every summer according to previous studies on the glacier (Uetake et al., 2010; Takeuchi et al., 2014). Additionally, the species causing this phenomenon are likely the same as those typically occurring in Arctic snowfields. The dominant algal cell, *Cd. nivalis*, has been widely reported in Arctic snowfields (Spijkerman et al., 2012; Takeuchi, 2013; Hisakawa et al., 2015; Lutz et al., 2016; Tanaka et al., 2016).

The red algal cells appear to have originated from windblown algal spores in the atmosphere, but they are not likely from the remaining snow of previous melting season. Algae growing on the snow surface are usually derived from spores transported by wind or animals from distant places (up to hundreds or kilometers) or from motile cells that migrated from the lower layers of the snowpack (Müller et al., 2001; Remias, 2012). The migration of motile cells in the snowpack requires solar radiation as well as liquid water (Hoham, 1980). However, photosynthetically active radiation can only penetrate to a depth of 1 m in wet

snowpacks (Curl et al., 1972). When snow algae appeared on the snow surface at the study sites, the previous summer surface was located deeper than 1 m from the present surface (229 cm and 110 cm at Sites-A and B, respectively). The depth appeared to be too great for these cells to migrate to the surface. Furthermore, there was superimposed ice over the last summer surface in the snowpack at Site-B when the algae appeared (Aoki et al., 2014). The superimposed ice layers seem to block algal migration to the surface. The lack of algal cell in the snow pit samples also suggest the algal cell are not derived from the lower snow layers. Therefore, the algal cells are unlikely to have originated from beneath the snow. Alternatively, algal cells might have been transported from the ground surface surrounding the glacier or from distant sources via atmosphere. Previous studies reported that mineral dust on glaciers in northwest and southwest Greenland is mainly supplied from local ground surfaces (e.g. moraine near the glacier), rather than the distant areas (Nagatsuka et al., 2014; 2016). Therefore, the algal spores, which have been washed out from the glacier and stayed on the ground, may be supplied with such dust around the glacier by wind.

Meteorological records suggest that the initiation of algal growth requires the air temperature to remain above 0°C for a certain period of time after the previous snowfall. The snow algae at both Sites-A and B appeared two days apart from each other. Prior to algal appearances, the hourly air temperature remained above 0°C for 94 h from day 175 at Site-A and for 136 h from day 176 at Site-B; there was no snowfall during this time at either site. The period from the last snowfall appears to be important in initiating snow algal growth, as fresh snow coverage inhibits photosynthesis of the snow algae under the snow. Additionally, snowmelt is required for the initiation of algal growth (Fukushima, 1963; Onuma et al., 2016). Snow algae on a snowpack in Japan has been reported to appear when air temperatures exceed 0°C for 24 h, which is likely the minimum requirement for initiating snow algal growth (Onuma et al., 2016). The duration was longer in this study than that which was observed in Japan. The longer duration may be due to a difference of algal species or weather conditions on this glacier. These results suggest that continuous melting for a minimum of 94 h is required for the initiation of algal growth on the Qaanaaq Glacier although further studies are necessary to determine the snow physical conditions for the initiation.

## 4.2 Approximation of the algal growth curve with a numerical model

In order to reproduce the observed algal growth with a numerical equation, we applied a logistic model that utilizes a general differential equation of microbial growth to the observed algal growth curve. An increase in microbial cells can simply be expressed by a differential equation known as the Malthusian model, which is defined by an initial cell concentration and algal growth rate (Lavoie et al., 2005). The Malthusian model is based on the assumptions that microbial abundance increases by cell division of all present cells at a constant rate, that there is no addition or removal of cells in the habitat, and that light, nutrients, and habitable space are unlimited. According to this model, the microbial growth curve is calculated as follows (Cui and Lawson, 1982):

$$X = X_0 e^{\mu(t-t_0)}, \tag{1}$$

where $X$ and $X_0$ are population densities of microbes at t and $t_0$, respectively, and $\mu$ is the growth rate of microbes in $t^{-1}$. The Malthusian model has been applied to observational microbial abundances in sea ice (Lavoie et al., 2005) and in snowfield

(Onuma et al., 2016). However, the algal abundance at Site-A did not significantly increase after late July, despite the air temperature remaining above 0°C and a lack of snowfall, indicating that the Malthusian model could not represent the algal growth curve on the surface snow of the Qaanaaq Glacier. The decreased growth rate observed on the glacier suggests that algal abundance has a limited capacity in this habitat. A logistic model is a microbial growth equation with a carrying capacity, and thus could represent the algal growth curve observed in this study. The temporal change of the logistic model is represented as follows (Cui and Lawson, 1982):

$$X = \frac{K}{1+\frac{K-X_0}{X_0}e^{\mu(t_0-t)}}, t = d - d_f, \tag{2}$$

where $K$ is the carrying capacity of algae in the snow surface (depth = 2 cm) and $t_0$ is the day of the first appearance of algae on the snow surface. Since snow algae can grow only on the melting snow surface, we assumed that algal growth was interrupted when snow surface temperature was below 0°C. Thus, $t$ represents the number of the days during which the mean temperature was above 0°C. This equation was fitted to the observational algal cell concentrations at Sites-A and B through Poisson regression. The observational data used are from the day of algal appearance (days 179 at Site-A and 181 at Site-B, $t_0$) through the last day of the study period (days 214 at Site-A and 215 at Site-B, $t_{max}$). This regression is based on the assumption that there is no inflow or outflow of algal cells on the snow surface. To fit Poisson regression to the observed algal cell concentrations, carrying capacity was assumed to be $3.5 \times 10^7$ cells m$^{-2}$ at both sites based on the observed maximum concentration of algal cells (day 214 at Site-A). Although it is uncertain whether the algal concentration at Site-A was the greatest on day 214 in this summer, the carrying capacity was likely around the value since the cell concentrations hardly increased from day 201 to 214 despite air temperatures remaining above 0°C. In contrast, the algal cell concentration at Site-B continued to increase significantly until day 215, suggesting that it did not reach the level of the carrying capacity at this site. The cell concentration would increase further after the day because the snow surface temperature calculated at Site-B kept above 0°C for a week. Although the carrying capacity possibly varies in different snow surfaces, it was assumed to be a same at Site-A and B in this study since they are on the same glacier.

No inflow of algal cells on the snow surface was assumed for this calculation because wind-delivery of algal cells appeared to be smaller compared with the abundance during the growth period. The initial concentration of algae on the surface ($3.1 \times 10^3$ cells m$^{-2}$ on day 179 at Site-A and 7.4 cells m$^{-2}$ on day 181 at Site-B), which is probably equivalent to the algal cells of the wind-delivery, was substantially smaller than the final concentration ($3.5 \times 10^7$ cells m$^{-2}$ on day 214 at Site-A and $5.0 \times 10^5$ cells m$^{-2}$ on day 215 at Site-B). Outcropping algal cells from the subsurface snow also appear to be insignificant because no algal cell was detected in all of the snow layers below the surface. The outflow of algal cells by melt water is also likely to be insignificant to affect the algal cell abundance on the snow surface since the algal cell concentration kept increasing during the study period.

Fitting the data to the model showed that the coefficients of determination in the regressions ($R^2$) were 0.64 and 0.96 at Sites-A and B, respectively, suggesting that the algal growth curve was reproduced well with the equation (Table. 1, Figs. 7 and 8). However, the confidence levels cannot calculate for the regression curve because the standard deviations for the

observed algal cell concentration increased over time, therefore, the uncertainty of the calculated algal abundance appears to become larger in the late of the melting season. The decline of algal cell concentration observed from days 201 to 208 at Site-A was not reproduced in the calculated growth curve. This is likely the reason for the lower $R^2$ value at Site-A. However, the calculated cell concentration ($3.4 \times 10^7$ cells m$^{-2}$) was consistent with the observed abundance ($3.5 \times 10^7$ cells m$^{-2}$) on day 214,

which was the day when algal cell concentration on surface snow was the greatest during the observational period; this suggests that the model can accurately reproduce the cell abundance in the order of magnitude and the timing when algal cell concentration reached the carrying capacity.

### 4.3 Factors affecting parameters of algal growth model

The growth rates ($\mu$) obtained from the regression of the algal growth curve did not significantly differ between the two sites, whereas the initial cell concentration ($X_0$) at the lower site was 100 times greater than that of the higher site (Table 1).

The difference in initial cell concentration ($X_0$) between the two sites was likely attributed to the abundance of initial algal spores supplied from the atmosphere. The initial cell concentration observed at Sites-A and B were $6.9 \times 10^2$ cells m$^{-2}$ and 6.3 cells m$^{-2}$, and the abundance of mineral particles on the snow surface was also significantly greater at Site-A (1.4 g m$^{-2}$) than

Site-B ($7.0 \times 10^{-3}$ g m$^{-2}$, Table 1). The sources of mineral particles on the Qaanaaq Glacier were mostly from local sediments, such as soil and moraine near the glacier (Nagatsuka et al., 2014). Therefore, the initial algal spores on surface snow are likely supplied with mineral particles by wind from surrounding ground surface. There is little information on the abundance of initial algal spores on snow surface. Estimation of initial algal concentrations from mineral particle abundance could be applied to this model in the other glaciers or snowfields since the abundance of mineral particles is widely available by observation,

the surface mass balance model on the Greenland Ice Sheet (Goelles et al., 2015), and atmospheric circulation models (Ginoux et al., 2001).

The growth rates ($\mu$) obtained from both sites were similar, suggesting that growth rates on the glacier are constant. The growth rates were 0.39 d$^{-1}$ and 0.42 d$^{-1}$ at Sites-A and B, respectively (Table 1). Although solar radiation might affect algal photosynthesis and thus their growth rate on the snowpack, the effect is unclear since there was no significant difference in the

July solar radiation among the two sites (218 vs. 244 W m$^{-2}$ for Sites-A and B, respectively; Student's $t$-test, $t = -0.99$, $P = 0.32 > 0.05$). According to the measurement of the growth rate of isolated snow algae, *Chloromonas nivalis* in the culture, it was 0.6 d$^{-1}$ at 18°C water (Leya et al., 2009), which is significantly greater than the growth rate of 0.39–0.42 d$^{-1}$ in the present study (Table 1). The lower growth rate in our study is likely due to the lower temperature of the algal habitat compared to the culture conditions, as growth rate of fresh water algae is dependent on water temperature (Eppley, 1972). The growth rate of snow

algae may also vary among algal species although further study is necessary.

The carrying capacity of snow algae may be determined by nutrient availability in the snowpack. The carrying capacity was estimated to be $3.5 \times 10^7$ cells m$^{-2}$ in this study, based on the observed growth curve (Table 1). Although there are no observational data on the carrying capacity of snow algae, it could be represented by the cell concentration of snow algal bloom

reported late in the melting season in previous studies. Cell concentrations of snow algal blooms reported previously in various geographical locations ranged from $5.1 \times 10^7$ to $7.5 \times 10^9$ cells m$^{-2}$ (Table 2), suggesting that carrying capacity varies among sites and might be determined by environmental conditions. There are two possible factors affecting carrying capacity: (1) the reduction in physical space available for microbial growth (i.e. the volume of liquid melt water, McKindsey et al., 2006) and

(2) the exhaustion of resources for the algae on the snow surface (Cui and Lawson, 1982). The maximum algal cell volume on surface snow (depth = 2 cm) at Site-A (day 214) was substantially smaller compared to the total volume of liquid-water in the surface snow obtained from the water content (0.19 mL m$^{-2}$ vs. 500 mL m$^{-2}$); this indicates that the physical space for algal growth in the habitat is not a factor affecting carrying capacity in this study. Nutrients such as nitrogen and phosphorus are essential elements for algal growth and are usually supplied from the atmosphere to the snow surface as aerosols. Phosphorus

supplied that is carried by wind to glaciers in the form of phosphate minerals easily becomes a limiting factor for algal growth compared to nitrogen, as the concentration of phosphorus was less than that of nitrogen in glaciers (Stibal et al., 2008). Addition of nutrients from outside likely increases snow algal abundance in the snowpacks reported by Ganey et al. (2017). Therefore, the carrying capacity may be determined by approximation using the relationship between observational abundance of algal cells and mineral dust on the snow surface although further study is necessary to substantiate this claim.

## 5 Conclusions

The temporal changes in snow algal abundance on snowpacks of the Qaanaaq Glacier in northwest Greenland were studied. Spherical algal cells filled with red pigment, which are likely *Cd. nivalis*, were dominant on the snowpack. The algal cells first appeared on the snow surface in late June when snow had melted and the air temperature remained above 0°C exceeded for

approximately 94 h. Algal abundance increased exponentially for a month, and then the growth rate decreased mid-July, even though the air temperatures remained above 0°C and no snowfall occurred. A logistic model was applied to the observed algal growth curves to reproduce abundance numerically using three parameters; the initial cell concentration ($X_0$), growth rate ($\mu$), and carrying capacity ($K$). The growth curves were reproduced with coefficients of determination ($R^2$) of 0.64 and 0.96 at the lower and higher sites, respectively. Our observational results suggest that the model parameters can be determined using the

environmental conditions (physical and chemical snow properties and meteorological conditions) of the glacier; thus, this logistic model has a potential to reproduce the snow algae on glaciers or ice sheet scale although further studies are necessary to determine the three parameters of the model. The parameters determined in this study were based on the observation of a single glacier and season, more observation data of the algal seasonal growth could reduce the uncertainty of the model. In order to validate and calibrate the model parameters in more extensive areas of the glacier or the ice sheet, satellite images

could be useful as recent study successfully quantified the red algal abundance on an icefield in Alaska (Ganey et al., 2017). Furthermore, it is important to understand the life cycle of snow algae including the process of atmospheric transportation of the algal spores and effect of nutrient dynamics within the surface snow. Our results demonstrate that a simple numerical

model could simulate the temporal variation in algal abundance on snow surface on a Greenlandic glacier. In future, coupling this algal model with a regional climate model in Greenland, such as the model proposed by Niwano et al. (2018), would enable us to estimate snow melting regarding the effect of algal blooming. In addition, the model would be useful to know the algal life cycle on the ice sheet.

**6 Acknowledgements**

We would thank to the filed campaign members of the SIGMA (Snow Impurity and Glacial Microbe effects on abrupt warming in the Arctic) project and GRENE (the Green Network of Excellence) Arctic Climate Change Research project in Greenland in 2014. We also thank two anonymous reviewers and an editor (Marco Tedesco) for helpful suggestions that greatly improved
this manuscript. This study was supported in part by Grant-in-Aids (23221004, 26247078, 26241020, 16H01772).

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

**Table 1: List of the parameters of a logistic model of snow algal growth obtained from the data of each study site.**

| Site | Initial cell Concentration $X_0$ (cells m$^{-2}$) | Growth rate $\mu$ (d$^{-1}$) | Carrying capacity $K$ (cells m$^{-2}$) |
|---|---|---|---|
| Site-A | $6.9 \times 10^2$ | 0.42 | $3.5 \times 10^7$ |
| Site-B | 6.3 | 0.39 | $3.5 \times 10^7$ |

**Table 2: List of maximum algal cell concentrations of red algal bloom reported from various snow fields over the world. Maximum algal cell concentrations per area (cells m$^{-2}$) were obtained by calculation from reported maximum algal cell concentrations per volume (cells mL$^{-1}$) assuming the snow density of granular snow to be 500 kg m$^{-3}$ and the depth of collected samples to be 0.02 m.**

| Study sites | Algal species | Maximum algal cell concentration (cells m$^{-2}$) | References |
|---|---|---|---|
| Oregon, USA | *Chlamydomonas nivalis* | $2.3 \times 10^9$ | Sutton, 1972 |
| Washington, USA | *Chloromonas brevispina* | $5.0 \times 10^9$ | Hoham et al., 1979 |
| Antarctica | *Mesotaenium berggrenii* | $1.0 \times 10^9$ | Ling and Seppelt, 1990 |
| Antarctica | *Chloromonas rubroleosa* | $2.0 \times 10^9$ | Ling and Seppelt, 1993 |
| California, USA | *Trochiscia americana* | $6.3 \times 10^8$ | Thomas, 1994 |
| Svalbard | *Chloromonas alpine* | $7.5 \times 10^9$ | Spijkerman et al., 2012 |
| Alaska, USA | *Chlyamidomonas nivalis* | $5.1 \times 10^7$ | Takeuchi, 2013 |
| SE-Greenland | *Chlyamidomonas nivalis* | $5.0 \times 10^8$ | Lutz et al., 2014 |

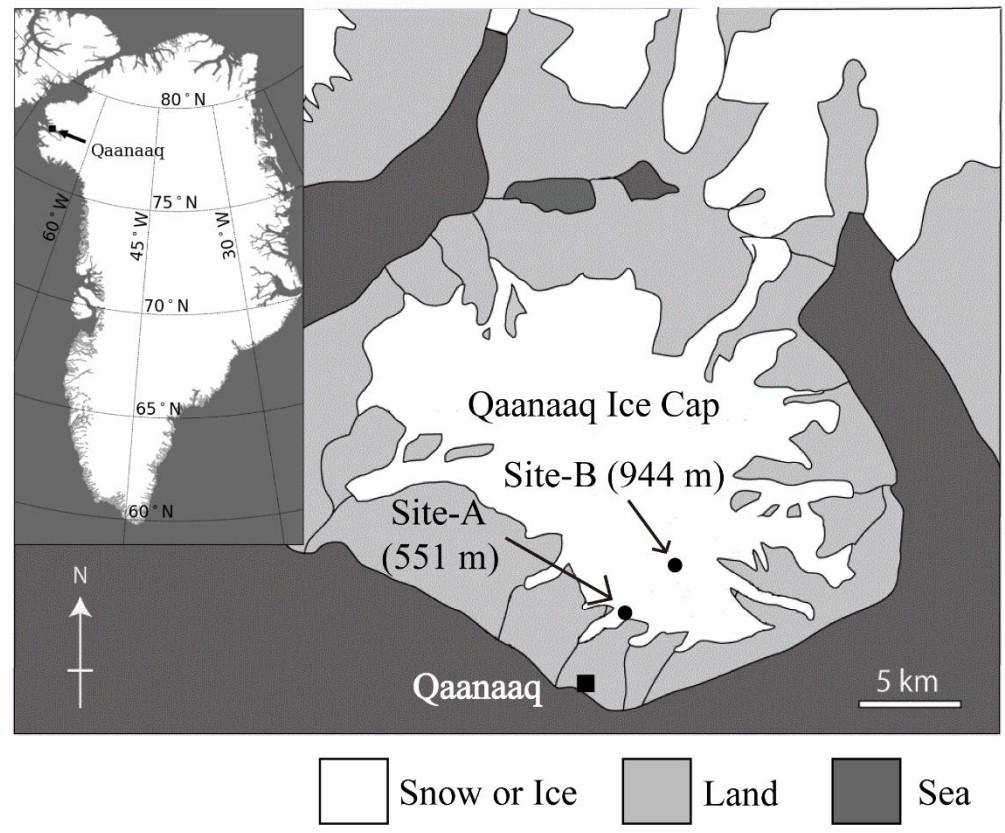

**Figure 1: A map of the Qaanaaq Ice Cap in northwest Greenland, showing the location of sampling sites on the glacier.**

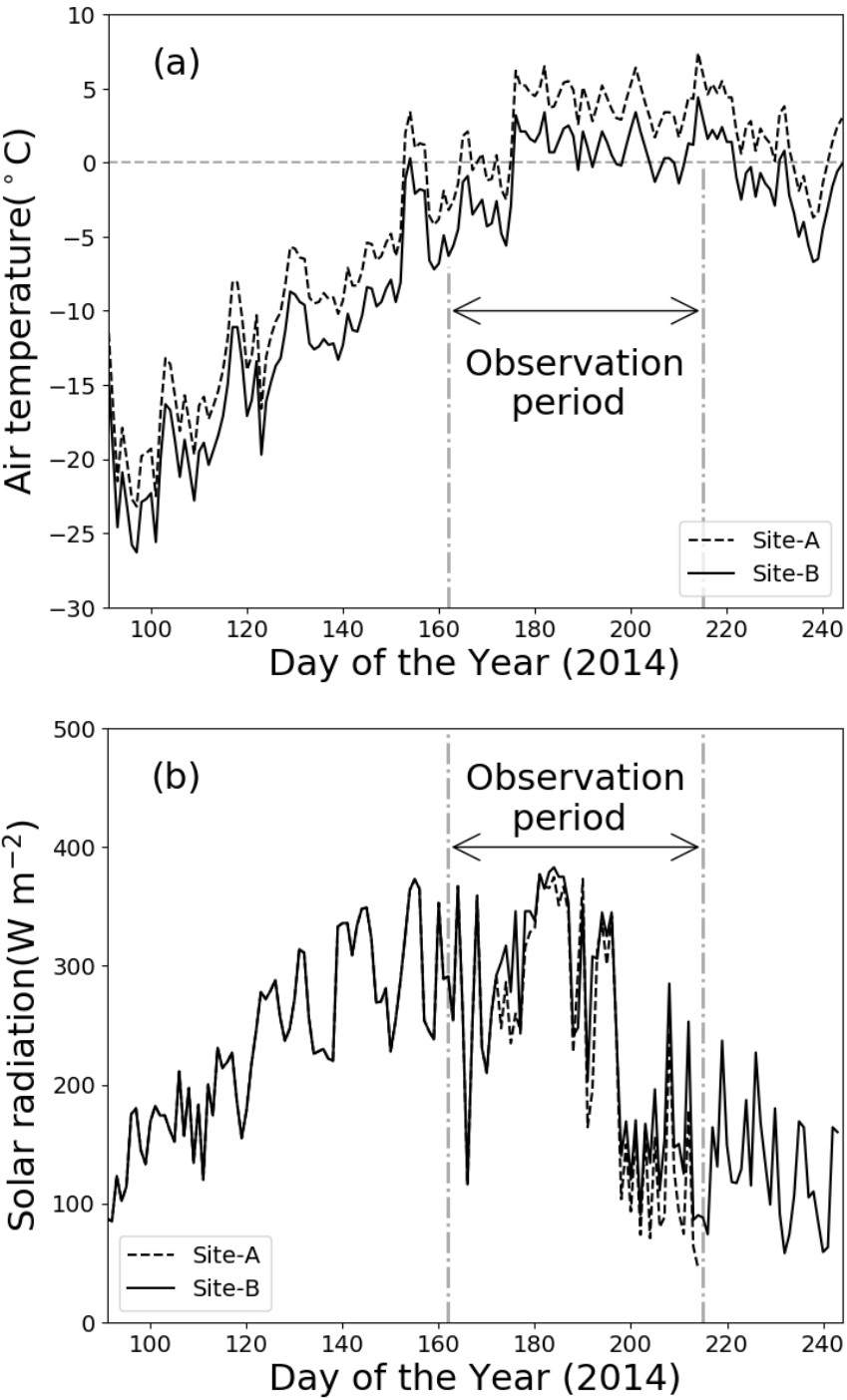

**Figure 2: Meteorological conditions at Sites-A and B from 1 April to 1 September 2014. (a) Daily mean air temperature, (b) solar radiation.**

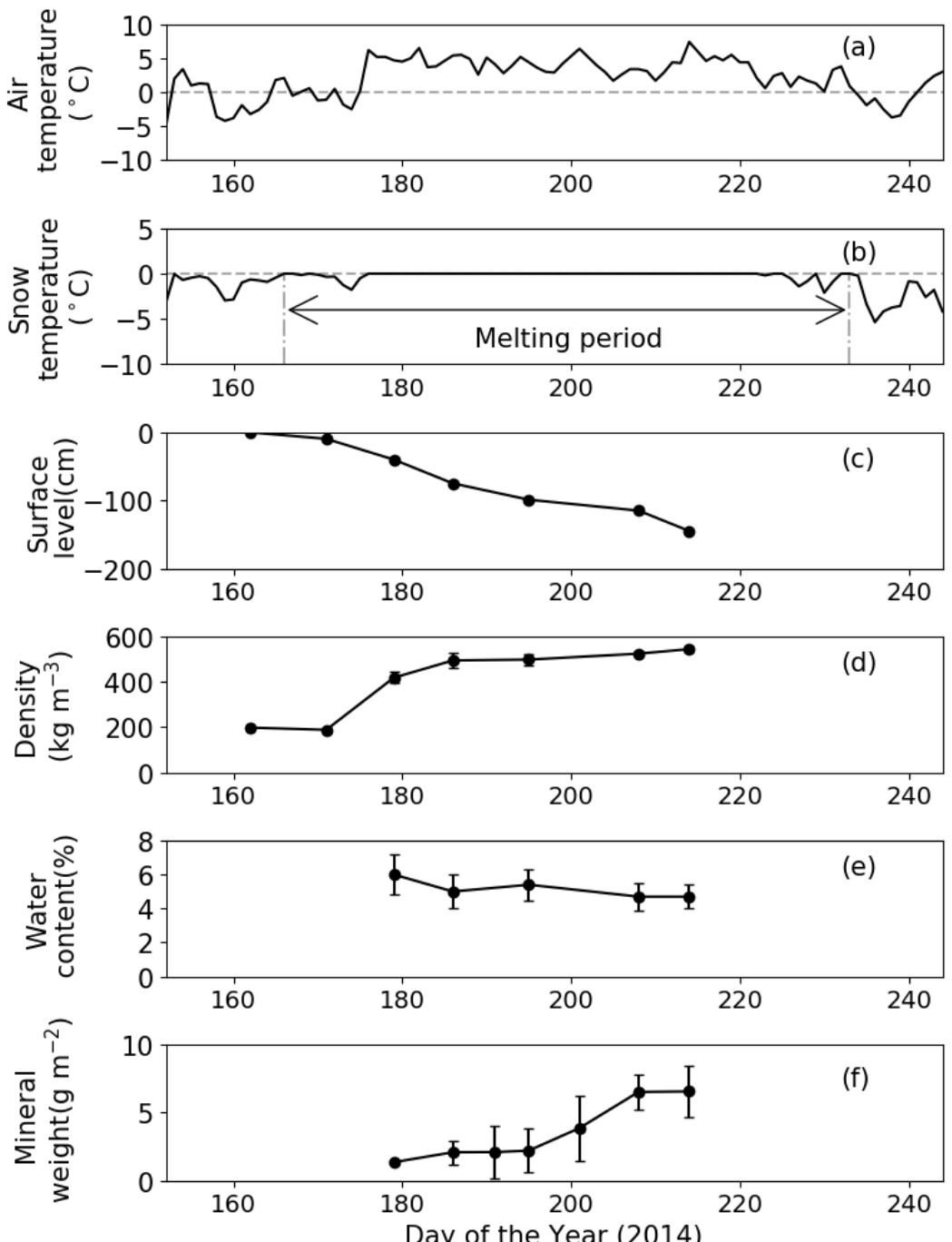

**Figure 3: Meteorological and physical conditions of surface snow at Site-A from 1 June to 1 September 2014. (a) Mean daily air temperature, (b) mean daily snow surface temperature calculated from observed downward and upward longwave radiant fluxes, (c) relative snow surface level at the site (0 cm on day 162), (d) snow density, (e) volumetric liquid-water content of snow, and (f) abundance of mineral particles. Melting period in (b) is defined as a period from first day until last day when mean daily snow surface temperature was 0°C from 1 June to 1 September 2014. Standard deviation shown by error bars.**

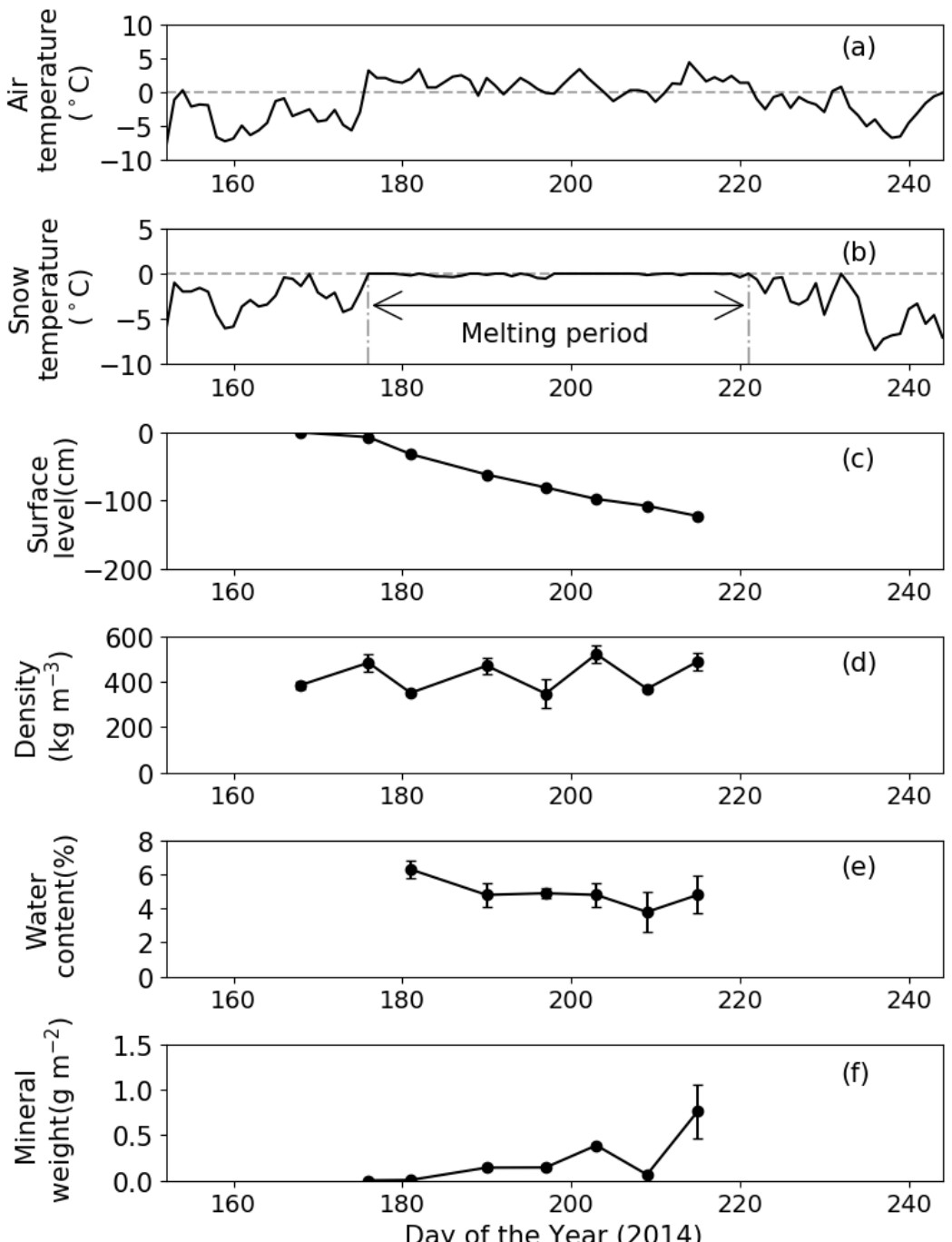

**Figure 4: Meteorological and physical conditions on surface snow at Site-B from 1 June to 1 September 2014. (a) Mean daily air temperature, (b) mean daily snow surface temperature calculated from observed downward and upward longwave radiant fluxes, (c) relative snow surface level at the site (0 cm on day 168), (d) snow density, (e) volumetric liquid-water content of snow, (f) abundance of mineral particles. Melting period in (b) is defined as a period from first day until last day when the daily mean snow surface temperature was 0°C from 1 June to 1 September 2014. Standard deviation shown by error bars.**

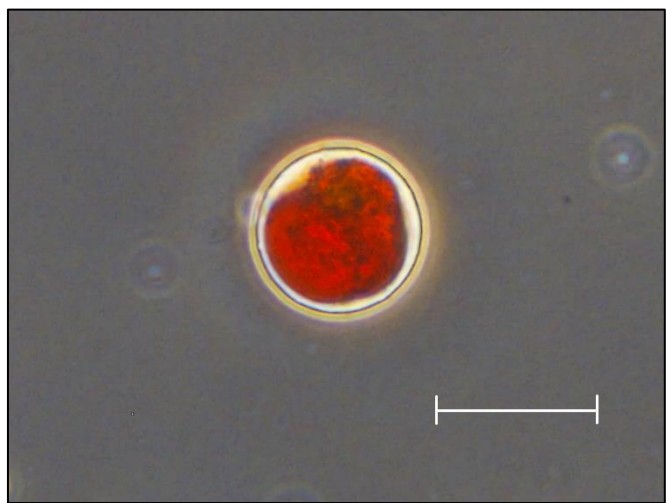

**Figure 5: Photograph of snow algal cell observed on the snow surface at Site-A. An oval red cell with secondary carotenoids, most likely mature spores of *Cd. nivalis*, which is the dominant species at both sites. Scale bar = 20 μm.**

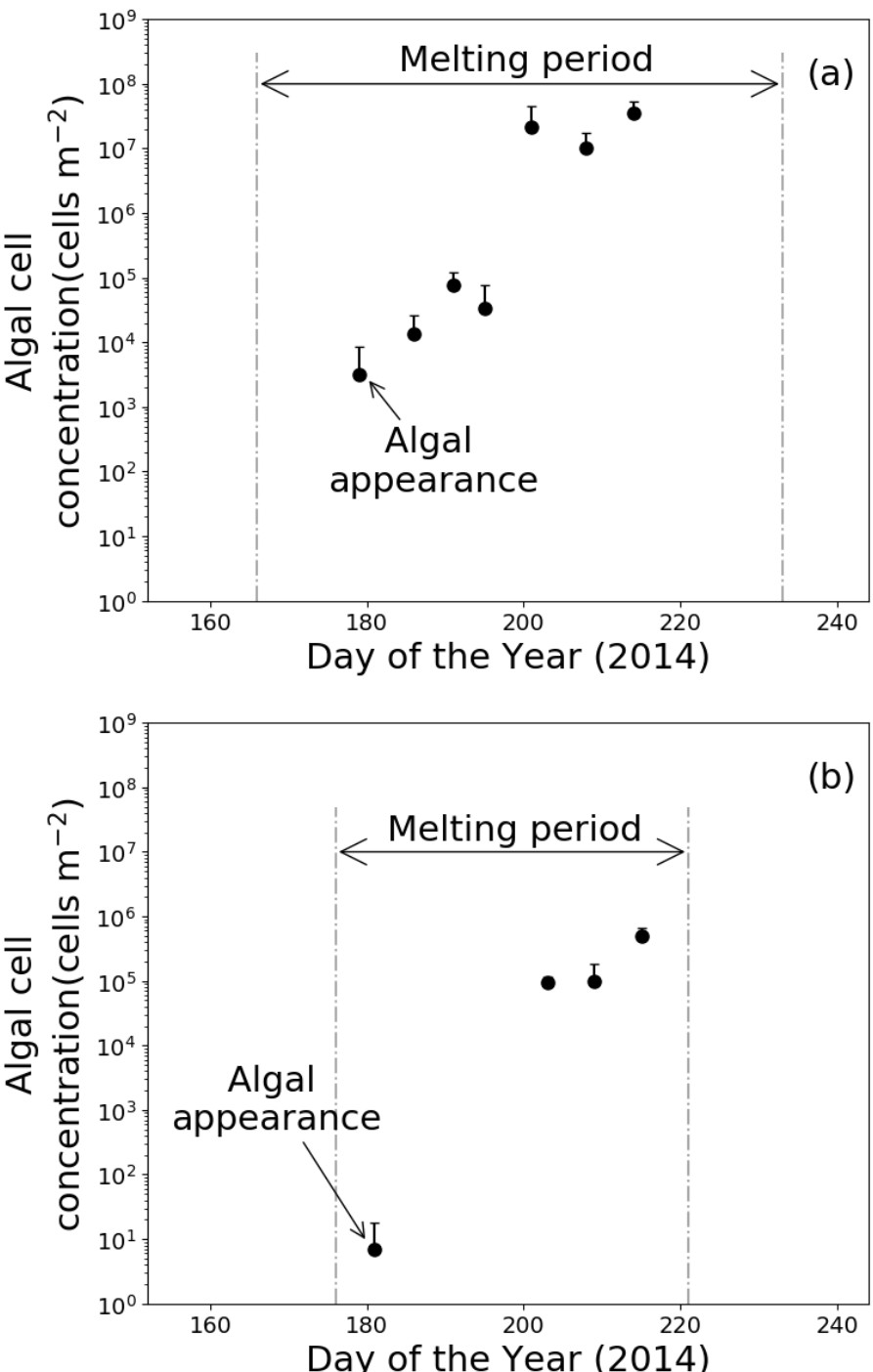

**Figure 6: Temporal changes in algal abundance on snow surface at (a) Site-A and (b) Site-B. Melting period in (a) and (b) indicated in Figs. 3 (b) and 4 (b), respectively. Standard deviation shown by error bars.**

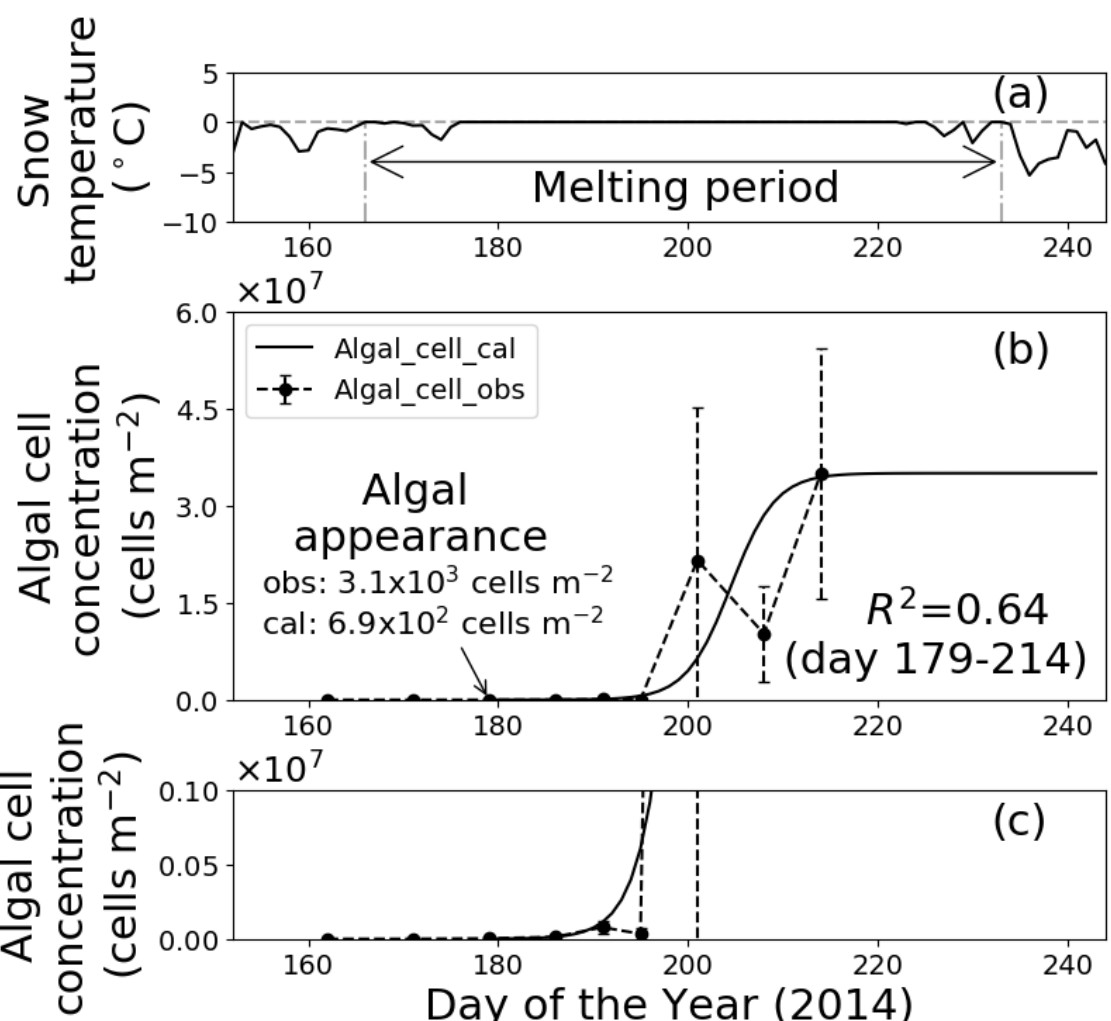

**Figure 7: Temporal changes in snow temperature and algal abundance on the surface at Site-A. (a) Mean daily snow surface temperature, (b) observed and calculated algal abundance, and (c) enlargement of the observed and calculated algal abundance between 0 and $1.0 \times 10^6$ cells m⁻². Surface snow temperature was calculated from observed downward and upward longwave radiant fluxes. Solid marks indicate observed algal abundance. Solid lines indicate algal abundance calculated from regression by logistic model. Standard deviation shown by error bars.**

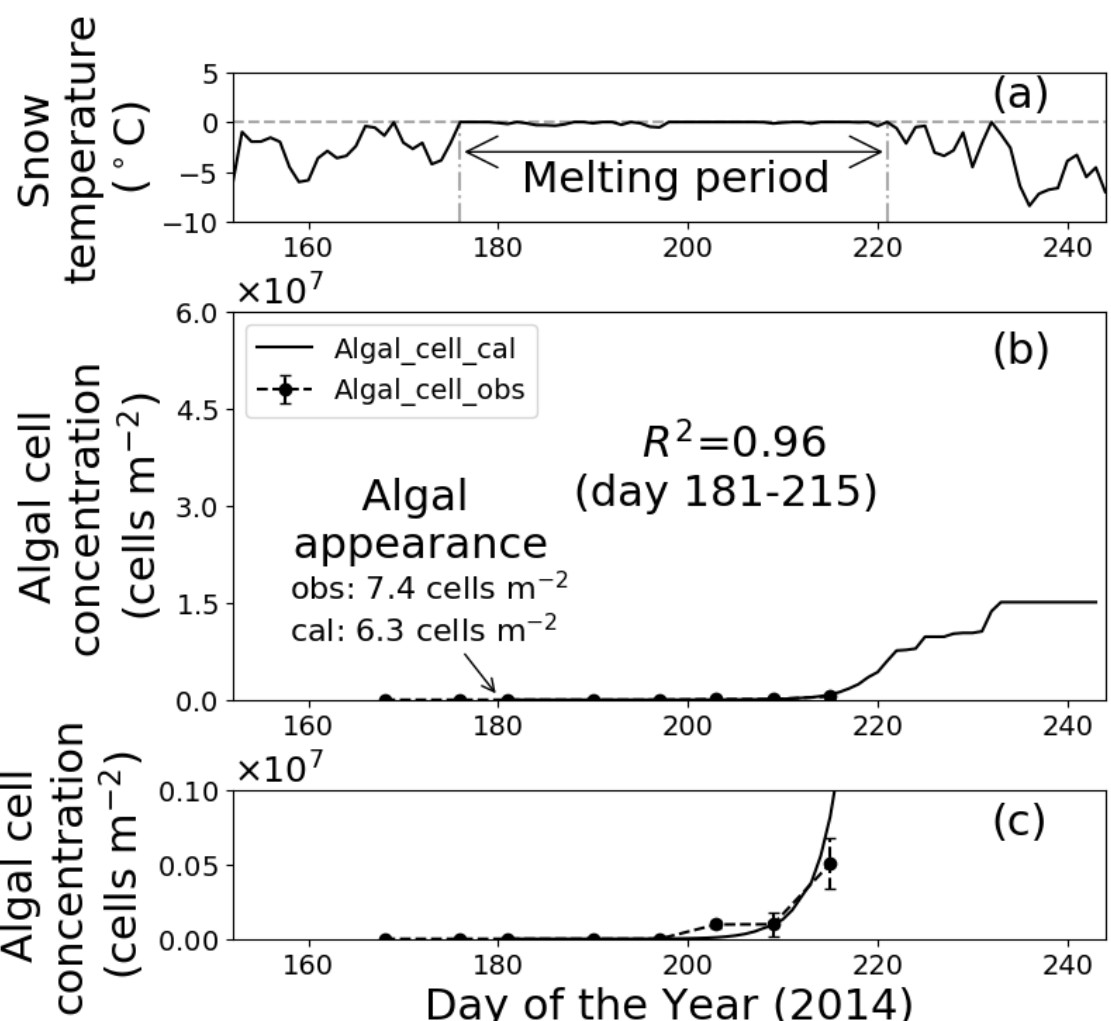

**Figure 8: Temporal changes in snow temperature and algal abundance on surface snow at Site-B. (a) Mean daily snow surface temperature, (b) observed and calculated algal abundance, and (c) enlargement of the observed and calculated algal abundance between 0 and $1.0 \times 10^6$ cells m$^{-2}$. Surface snow temperature was calculated from observed downward and upward longwave radiant fluxes. Solid marks indicate observed algal abundance. Solid lines indicate algal abundance calculated from regression by logistic model. Standard deviation shown by error bars.**