# Peer review of "Observations and modelling of algal growth on a snowpack in northwest Greenland"

_The Cryosphere, 2017_

## Referee Comment (RC1) · Anonymous Referee #1 · 16 Dec 2017

Dear Authors,

The submitted manuscript aims to address a worthwhile gap in our knowledge of snow algae – namely the ability to model the growth of algal blooms over a season. The paper combines empirical observations with a simple growth model, building upon previous work by the same group that applied a Malthusian model to explain the population dynamics of the snow algal bloom by adding a carrying capacity. I enjoyed reading the paper and appreciated the careful field measurements; however, I have two queries for the authors, along with some minor typographical corrections.

1. The model assumes no input or removal of cells, indicating that the population dynamics are dominated by in situ cell proliferation. However, no mention is made of the effects of scavenging by melting snow. The algae are likely to become concentrated

onto the snow surface as the snowpack melts, and if this is the case then the authors would have measured an increase in surface biomass simply due to the concentrating effects of snow melt. Can the authors show that this was not the case? Similarly, it is hard to envisage zero cell export from the system. I suspect the snowpack algal population dynamics to be the result of in situ growth, scavenging by snow melt, export in meltwater and a small contribution by wind-delivery. Can the authors provide any data or observations to support all these factors being negligible apart from in situ growth?

2. I think the authors overstate the utility of the model. As it stands the manuscript shows that a logistic modelling approach is appropriate for predicting the population dynamics of snow algae for specific field sites; however, insufficient information is provided to enable the application of the model elsewhere. The fact that between these two sites the coefficient of determination for the model applied to observations varies between 0.64 and 0.96 suggests to me that additional site specific factors influence the growth rate. K is, as the authors discuss, dependent upon the nutrient availability and available space, but no quantitative link is drawn between either variable and the value of K. The slope of the model is assumed to be entirely dependent upon time, but it sees unlikely that this assumption would often be satisfied. Presumably, the rate of growth is in reality affected by many more variables. For example the study by Onuma et al. (2016) found algal blooms to initiate just 24 hours after melting began which the authors attribute to either a different algal species (can you confirm this with observations?) or 'weather conditions'. This suggests that the growth rate varies between sites and cannot be assumed constant. As I understand it, weather conditions are precisely what you are trying to analyse as potential drivers of algal growth in this paper, so it seems contradictory to invoke them as an additional source of uncertainty. It seems likely that the initiation and growth rate of snow algal blooms are the result of a combination of meteorological factors plus site specific variations such as nutrient availability in the snowpack that may well be difficult to unpick, limiting the predictive capability of the model on other snowpacks. Can the authors provide any more data or discussion

to support the applicability of the model?

Specific Comments:

pg 1 line 23: change 'bloom' to 'blooms', change 'changes' to 'change' pg 1 line 24: change 'bloom' to 'blooms' pg 2 line 20: delete 'works' pg 2 line 26: these references are a mixture of ice algae and snow algae. Ice algae has been suggested to enhance ablative losses to the GrIS by reducing albedo (see more recent papers by Tedstone et al. 2017 and Stibal et al. 2017). To my knowledge, evidence for accelerated snow line retreat due to snow algal blooms has not yet been presented for Greenland. I suggest rewording this sentence accordingly or providing specific references for accelerated snow line retreat. Pg2 line 28: The Lutz studies were limited to the visible wavelengths and were not really measuring albedo, but a proxy. Better to say that they showed algae can modulate snow reflectance in the visible wavelengths. Pg3 line 5: superscript km^2 pg4 line 4: normalising to area presumably requires that the algae only inhabit an extremely thin layer on the upper surface of the snow – other studies (e.g. Thomas et al., 1979; Hodson et al., 2017) indicate that subsurface red algae can exist. Are you confident that the algae were confined to the upper surface? Is this supported by your observations? Pg6 line 21: change 'active radiation' to 'photosynthetically active radiation' pg 7 line 24: delete 'can' pg 8 line 33: statistical significance should be supported by test name and values. pg8 line 28: change 'snowfileds' to 'snowfields'

References

Hodson, A. J., A. Nowak, J. Cook, M. Sabacka, E. S. Wharfe, D. A. Pearce, P. Convey, and G. Vieira (2017), Microbes influence the biogeochemical and optical properties of maritime Antarctic snow, J. Geophys. Res. Biogeosci., 122, 1456–1470, doi:10.1002/2016JG003694.

Leya, T., Rahn, A., Lütz, C., and Remias, D., Response of arctic snow and permafrost algae to high light and nitrogen stress by changes in pigment composition and applied aspects for biotechnology, FEMS Microbiol. Ecol., 67, 432–443, doi: 10.1111/j.1574-

6941.2008.00641.x, 2009.

Onuma, Y., Takeuchi, N., and Takeuchi, Y.: Temporal changes in snow algal abundance on surface snow in Tohkamachi, Japan, Bull. Glaciol. Res., 34, 21-31, doi:10.5331/bgr.16A02, 2016

Stibal, M., Box, J. E., Cameron, K. A., Langen, P. L., Yallop, M. L., Mottram, R. H., . . . Ahlstrøm, A. P. (2017). Algae drive enhanced darkening of bare ice on the Greenland ice sheet. Geophysical Research Letters, 44. https://doi.org/10.1002/2017GL075958

Tedstone, A. J., Bamber, J. L., Cook, J. M., Williamson, C. J., Fettweis, X., Hodson, A. J., and Tranter, M.: Dark ice dynamics of the south-west Greenland Ice Sheet, The Cryosphere, 11, 2491-2506, https://doi.org/10.5194/tc-11-2491-2017, 2017.

Thomas W. H. 1972 — Observations on snow algae in California — J. Phycol.. 8: 1—9.

---

## Referee Comment (RC2) · Anonymous Referee #2 · 23 Feb 2018

Summary:

The manuscript tries to understand the link between snow algae growth and its relationship with albedo and ablation rates. The authors use field observations to monitor changes in snow algae over the melt season and apply these results to a model to simulate algae blooms. This research is relevant to further understanding of how algae abundance on snowpacks evolution over a season and what effects they have on surface albedo and melt rates. While I believe this work is relevant to the community, I think the manuscript could be written in a more compelling way, with greater connections and applicability to the Greenland ice sheet. And, while the model serves its purpose for this study, I think its functionality should not be overstated. And, the linkage with snow algae to surface albedo and melt rates is not made in the manuscript. I think greater emphasis and connection with albedo and melting should be added. And, what these observation and modeling efforts mean for implementation into regional climate models.

Major Comments:

1. How representative is the model for use in other regions, beyond a glacier ice cap? Can the numerical model be feasibly used elsewhere on the ice sheet?
2. What are the larger impacts of this study? I think the authors should discuss this further and link the field and modeling study to broader application and regions of the Greenland ice sheet.
3. There appears to be large uncertainty associated with the algae cell observations (Fig. 7b). How can the authors argue that a good fit is achieved between the field and modeled algal cell concentration? There needs to be further discussion on the utility of the logistic model as well as its deficiencies. How can we improve the model? What data and additional variables are needed? And, what is the greater link to surface albedo and melting?

Specific Comments:

1. Pg. 6 line 2: Change to '$3.1*10^3$ cells m^-2'. And, again on line 4.
2. Pg. 6 line 17-18: What evidence do you have to validate that the red algal cells originate from windblown spores? Is there a way to verify this further and possible local sources (eg. nearby tundra)?
3. Pg. 7 line 5-6: reword sentence structure.
4. Pg. 7 Equations 1 and 2: These equations may be better placed in the Methods section.

5. Pg. 8 line 2-3: are these numbers correct? The text states the initial concentration was substantially smaller than the final concentration. Check the concentration numbers.
6. Pg. 8 line 4-5: Why aren't the authors using two separate carrying capacities for Site-A and Site-B, if they have different maximum concentrations of algal cells?
7. Pg. 8 line 21-22: The text of 100 times more at Site-A than Site-B is redundant to the previous few lines of text.
8. Pg. 24 Fig. 7b and c: Error bounds are needed for the logistic model (solid) line. Similarly, for Fig. 8b and c.

---

## Author Comment (AC1) · 27 Mar 2018

The submitted manuscript aims to address a worthwhile gap in our knowledge of snow algae – namely the ability to model the growth of algal blooms over a season. The paper combines empirical observations with a simple growth model, building upon previous work by the same group that applied a Malthusian model to explain the population dynamics of the snow algal bloom by adding a carrying capacity. I enjoyed reading the paper and appreciated the careful field measurements; however, I have two queries for the authors, along with some minor typographical corrections.

We would appreciate very much a number of constructive comments. We also glad in being able to hear that you enjoyed reading our paper. Our responses (blue text) to each the reviewer's comment (in black text) were described as follows. We also uploaded manuscript, which was revised with yellow marker as suggested, on the discussion board.

Major Comments:

1. The model assumes no input or removal of cells, indicating that the population dynamics are dominated by in situ cell proliferation. However, no mention is made of the effects of scavenging by melting snow. The algae are likely to become concentrated onto the snow surface as the snowpack melts, and if this is the case then the authors would have measured an increase in surface biomass simply due to the concentrating effects of snow melt. Can the authors show that this was not the case? Similarly, it is hard to envisage zero cell export from the system. I suspect the snowpack algal population dynamics to be the result of in situ growth, scavenging by snow melt, export in meltwater and a small contribution by wind-delivery. Can the authors provide any data or observations to support all these factors being negligible apart from in situ growth?

As the reviewer pointed out, algal cell abundance is possibly increased by the effects of scavenging or wind-delivery and reduced by the effect of melt water. We conducted additional analysis to discuss the effect of scavenging on algal growth in snowpacks (please see the Figures S1 and S2 in this response letter). The results show vertical profiles in algal cell concentration, snow density and snow temperature in snowpit of the study sites before the first appearance of snow algae on the surface. The vertical profiles showed that all of the snow layers in the study sites didn't include any algal cells, indicating that a scavenging by snow melt don't affects a temporal change in algal abundance. The effects of wind-delivery appeared to be small enough to calculate algal growth because the initial concentration of algae on the surface, which was likely due to wind in the study sites, was substantially smaller than the final concentration (Site-A: $3.1 \times 10^3$ cells m$^{-2}$ vs. $3.5 \times 10^7$ cells m$^{-2}$, Site-B: 7.4

cells m$^{-2}$ vs. $5.0 \times 10^5$ cells m$^{-2}$). The movement of algal cells by melt water also appeared to be small enough to calculate algal growth because algal cell concentrations in the study sites gradually increased with snow melting except for the period from day 201 to 214 at Site-A. Therefore, we simulated the temporal changes in algal cell concentration using logistic model based on the assumption that there is no inflow or outflow of algal cells on the snow surface. We have revised the manuscript to discuss the effects of scavenging, wind-delivery and melt water on a temporal variation in algal abundance (pg 4 lines from 4 to 6, pg 6 lines from 10 to 11 and pg 8 lines from 17 to 26). And, we have added the snowpit result to the discussion about the origination of algal cell (pg 7 lines from 6 to 7).

2. I think the authors overstate the utility of the model. As it stands the manuscript shows that a logistic modelling approach is appropriate for predicting the population dynamics of snow algae for specific field sites; however, insufficient information is provided to enable the application of the model elsewhere. The fact that between these two sites the coefficient of determination for the model applied to observations varies between 0.64 and 0.96 suggests to me that additional site specific factors influence the growth rate. K is, as the authors discuss, dependent upon the nutrient availability and available space, but no quantitative link is drawn between either variable and the value of K. The slope of the model is assumed to be entirely dependent upon time, but it sees unlikely that this assumption would often be satisfied. Presumably, the rate of growth is in reality affected by many more variables. For example the study by Onuma et al. (2016) found algal blooms to initiate just 24 hours after melting began which the authors attribute to either a different algal species (can you confirm this with observations?) or 'weather conditions'. This suggests that the growth rate varies between sites and cannot be assumed constant. As I understand it, weather conditions are precisely what you are trying to analyse as potential drivers of algal growth in this paper, so it seems contradictory to invoke them as an additional source of uncertainty. It seems likely that the initiation and growth rate of snow algal blooms are the result of a combination of meteorological factors plus site specific variations such as nutrient availability in the snowpack that may well be difficult to unpick, limiting the predictive capability of the model on other snowpacks. Can the authors provide any more data or discussion to support the applicability of the model?

Generally, the Michaelis-Menten equation has been used to control a growth rate of microbial cell using a nutrient concentration and coupled with the Malthusian model for utility reproduction of algal growth in lakes or oceans (e.g. Lavoie et al., 2005). However, we consider that it is difficult to use Michaelis-Menten equation, which requires a temporal

change in nutrient concentration related to algal growth, for simulation of algal growth rate on a glacier now because it is hard to analyze a temporal change in nutrient concentration in various glaciers. On the other hand, modeling using a carrying capacity of a logistic model has the advantage of that data of nutrient concentration in snow aren't needed to simulate algal growth. Carrying capacities in various snowfields is likely to be obtained by approximation between observational maximum algal abundance and mineral dust weight because nutrients, which are required for algal growth, are supplied on glacier with mineral dust as described in the manuscript (pg 10 line 13). For these reasons, we used a logistic model to simulate algal growth in this study. As the reviewer pointed out, there probably is an insufficient information for applying of logistic model to various fields. In future, a logistic model should be validated or improved on the basis of biological process in order to accurately reproduce a temporal variation in algal cell abundance and apply the model to various snowfields. We have revised the manuscript about the utility of a logistic model to apply to snowfields worldwide and the discussion of carrying capacity as suggested (pg 10 lines from 18 to 20 and pg 11 lines from 1 to 6).

Specific Comments:
1.  pg 1 line 23: change 'bloom' to 'blooms', change 'changes' to 'change'
The words have been corrected (pg 1 line 23).

2.  pg 1 line 24: change 'bloom' to 'blooms'
The word has been corrected (pg 1 line 24).

3.  pg 2 line 20: delete 'works'
The word has been deleted (pg 2 line 20).

4.  pg 2 line 26: these references are a mixture of ice algae and snow algae. Ice algae has been suggested to enhance ablative losses to the GrIS by reducing albedo (see more recent papers by Tedstone et al. 2017 and Stibal et al. 2017). To my knowledge, evidence for accelerated snow line retreat due to snow algal blooms has not yet been presented for Greenland. I suggest rewording this sentence accordingly or providing specific references for accelerated snow line retreat.
The sentence has been revised as suggested (pg 2 lines from 26 to 29). We have added the explanation about reduction of surface ice albedo due to ice algal bloom and revised the explanation about accelerated of snow melting due to snow algal bloom as suggested. The following reference has been added at pg2 line 27.

Stibal, M., Box, J. E., Cameron, K. A., Langen, P. L., Yallop, M. L., Mottram, R. H., ··· Ahlstrøm, A. P.: Algae drive enhanced darkening of bare ice on the Greenland ice sheet, Geophysical Research Letters, 44, doi: 10.1002/2017GL075958, 2017.

Tedstone, A. J., Bamber, J. L., Cook, J. M., Williamson, C. J., Fettweis, X., Hodson, A. J., and Tranter, M.: Dark ice dynamics of the south-west Greenland Ice Sheet, The Cryosphere, 11, 2491-2506, doi: 10.5194/tc-11-2491-2017, 2017.

5. Pg2 line 28: The Lutz studies were limited to the visible wavelengths and were not really measuring albedo, but a proxy. Better to say that they showed algae can modulate snow reflectance in the visible wavelengths.

The sentence has been revised as suggested (pg2 line 29).

6. Pg3 line 5: superscript km^2

The word has been corrected (pg 3 line 10).

7. pg4 line 4: normalising to area presumably requires that the algae only inhabit an extremely thin layer on the upper surface of the snow – other studies (e.g. Thomas et al., 1979; Hodson et al., 2017) indicate that subsurface red algae can exist. Are you confident that the algae were confined to the upper surface? Is this supported by your observations?

As the response to major comment 1, the vertical profiles in algal cell concentration in snowpack of the study sites show that there were no algal cells in snowpack before snow algae first appear in surface snow (Figures S1 and S2). In addition, snow algal cells are likely to be supplied from the atmosphere on surface snow in the study sites. These results suggest that snow algal cells concentrated to surface snow although a part of the algal cells may be flowed to subsurface snow by the melt water. We have revised the manuscript to discuss the effect of algal cells in subsurface snow on algal growth in surface snow (pg 8 lines from 21 to 24).

8. Pg6 line 21: change 'active radiation' to 'photosynthetically active radiation'

The words have been changed (pg 7 line 1).

9. pg 7 line 24: delete 'can'

The word has been deleted (pg 8 line 11).

10. pg 8 line 33: statistical significance should be supported by test name and values.
We have added a result of statistical test (Student's $t$-test) to the sentence (pg 9 lines from 29 to 30).

11. pg8 line 28: change 'snowfileds' to 'snowfields'
The word has been corrected (pg 9 line 23).

[Figure]

Figure S1. Vertical profiles of snow algal abundance and physical properties in a snow pit on day 162 at Site-A. (a) algal cell concentration, (b) snow density, (c) snow temperature in snow. Standard deviation shown by error bars.

[Figure]

Figure S2. Vertical profiles of snow algal abundance and physical properties in a snow pit on day 168 at Site-B. (a) algal cell concentration, (b) snow density, (c) snow temperature in snow. Standard deviation shown by error bars.

---

## Author Comment (AC2) · 27 Mar 2018

Summary:

The manuscript tries to understand the link between snow algae growth and its relationship with albedo and ablation rates. The authors use field observations to monitor changes in snow algae over the melt season and apply these results to a model to simulate algae blooms. This research is relevant to further understanding of how algae abundance on snowpacks evolution over a season and what effects they have on surface albedo and melt rates. While I believe this work is relevant to the community, I think the manuscript could be written in a more compelling way, with greater connections and applicability to the Greenland ice sheet. And, while the model serves its purpose for this study, I think its functionality should not be overstated. And, the linkage with snow algae to surface albedo and melt rates is not made in the manuscript. I think greater emphasis and connection with albedo and melting should be added. And, what these observation and modeling efforts mean for implementation into regional climate models.

We would appreciate very much a number of constructive comments. We also appreciate that you evaluated our approach to further understanding of temporal change in algal abundance on snowpacks although our manuscript needs more revising. Our responses (blue text) to each the reviewer's comment (in black text) were described as follows. We also uploaded manuscript, which was revised with yellow marker as suggested, on the discussion board.

Major Comments:

1. How representative is the model for use in other regions, beyond a glacier ice cap? Can the numerical model be feasibly used elsewhere on the ice sheet?

Logistic model requires three parameters, which are initial cell concentration, growth rate and carrying capacity, to calculate temporal change in algal cell abundance, so we consider that it is important to decide the parameters in various snow fields for reproduction of algal abundance in various regions. Although there is a little information of these parameters in other regions, the initial cell concentration and carrying capacity is likely to be related to mineral particle weight and snow chemical properties in our study, respectively. The growth rate of snow algae may be decided to constant value each species because the growth rates in two study sites were close to each other. The factors effect on the model parameters will be studied for improvement of the model. In addition, we'll validate and calibrate the model parameters in various fields in the future. Observational data of snow algal abundance for the validation and calibration will be collected from field or satellite observation. We consider that we may be able to validate and calibrate the model parameters in various fields because the method to estimate algal cell abundance on surface snow using Landsat8 images have been presented (Ganey et al., 2017). We have added an explanation about future task to

reproduce algal abundance on other snow fields (from Pg 11 lines from 1 to 3).

2. What are the larger impacts of this study? I think the authors should discuss this further and link the field and modeling study to broader application and regions of the Greenland ice sheet.

We consider that reproduction of temporal change in snow algal abundance using a numerical model is important to estimate mass balance of Greenland ice sheet more accurately with modeling because blooming of red snow algae can reduce snow albedo. For the estimation of the algal effect on snow albedo in Greenland ice sheet, logistic model should be coupled with a regional snow physical model (e.g. Niwano et al., 2018) to simulate snow physical properties including snow albedo in future. Also, a numerical model for algal growth may supply the useful information for study about life cycle of snow algae based on field observation by glacial biologist. For example, glacial biologist may able to project the timing of algal blooming from the simulation result of the algal growth model. As reviewer pointed out, our discussion was insufficient about potential of contribution to other modeling or field observation. We have added the explanation to the manuscript (Pg 11 lines from 6 to 9). The following reference has been added at Pg 11 line 6.

Niwano, M., Aoki, T., Hashimoto, A., Matoba, S., Yamaguchi, S., Tanikawa, T., Fujita, K., Tsushima, A., Iizuka, Y., Shimada, R. and Hori, M.: NHM–SMAP: spatially and temporally high-resolution nonhydrostatic atmospheric model coupled with detailed snow process model for Greenland Ice Sheet, The Cryosphere, 12, 635–655, https://doi.org/10.5194/tc-12-635-2018, 2018.

3. There appears to be large uncertainty associated with the algae cell observations (Fig. 7b). How can the authors argue that a good fit is achieved between the field and modeled algal cell concentration? There needs to be further discussion on the utility of the logistic model as well as its deficiencies. How can we improve the model? What data and additional variables are needed? And, what is the greater link to surface albedo and melting?

In this study, we aim to propose a simple numerical model for reproduction of algal growth in snowpacks. As reviewer pointed out, although further improvement of algal growth model is needed to reproduce a temporal change in algal abundance on snowpack more accurately, our results suggest that logistic model can simulate the timing of algal blooming. We didn't propose more complex model, which can simulate algal abundance including other factors affecting algal growth (e.g. movement of algal cells in snowpack), and estimate the effect of algal growth on snow albedo. However, we'll try to simulate a temporal change in algal abundance and snow albedo using a coupled logistic model with a snow physical model (e.g. Aoki et al., 2011; Niwano et al., 2012) in

the future. For example, we consider that the coupled logistic model tries to simulate algal abundance including the effect of the cells outflow by melt water on algal growth or snow albedo including the effect of algal blooming on light absorption in snow. Temporal changes in algal abundance and physical properties each snow layer should be needed to validate and calibrate the coupled algal growth model. We have revised the manuscript to reflect reviewer's comments (from Pg 2 line 32 to Pg 3 line 1 and Pg 11 lines from 6 to 9). The following reference has been added at Pg 2 line 33.

Aoki, T., Kuchiki, K., Niwano, M., Kodama, Y., Hosaka, M., and Tanaka, T.: Physically based snow albedo model for calculating broadband albedos and the solar heating profile in snowpack for general circulation models, J. Geophys. Res., 116, D11114, https://doi.org/10.1029/2010JD015507, 2011.

Niwano, M., Aoki, T., Kuchiki, K., Hosaka, M., and Kodama, Y.: Snow Metamorphism and Albedo Process (SMAP) model for climate studies: Model validation using meteorological and snow impurity data measured at Sapporo, Japan, J. Geophys. Res., 117, F03008, https://doi.org/10.1029/2011JF002239, 2012.

Specific Comments:
1.  Pg. 6 line 2: Change to '3.1*10^3 cells m^-2'. And, again on line 4.
The words have been corrected (Pg 6 lines 13 and 15).

2.  Pg. 6 line 17-18: What evidence do you have to validate that the red algal cells originate from windblown spores? Is there a way to verify this further and possible local sources (eg. nearby tundra)?
Our result and previous studies suggest that the algal cell spores in the study sites are supplied with mineral particles from moraine near Qaanaaq Glacier. As described in the manuscript, initial cell concentration is likely to be related to mineral particle weight. Previous studies reported that mineral dust on glaciers in northwest and southwest Greenland is likely to be supplied from local sediments (e.g. moraine near the glacier), rather than the distant areas (Nagatsuka et al., 2014; 2016). The algal cell spores may be on the moraine near the glacier because the algal cells are probably flowed to outside (e.g. moraine) of the glacier when snowpack including algal cells was disappeared. Therefore, origination of the algal cell spores may be moraine near the glacier. We have revised the manuscript to discuss about origination of algal cell spores more (Pg 7 lines from 9 to 13). The following reference has been added at Pg 7 line 11.

Nagatsuka, N., Takeuchi, N., Uetake, J. and Shimada, R.: Mineralogical composition of cryoconite on glaciers in northwest Greenland. Bull. Glaciol. Res., 32, 107–114, doi:10.5331/bgr.32.107, 2014.

Nagatsuka, N., Takeuchi, N., Uetake, J., Shimada, R., Onuma, Y., Tanaka, S. and Nakano, T.: Variations in Sr and Nd isotopic ratios of mineral particles in cryoconite in western Greenland. Front. Earth Sci., 4, 93, doi: 10.3389/feart. 2016.00093, 2016.

3. Pg. 7 line 5-6: reword sentence structure.

The sentence has been revised (Pg 7 lines from 23 to 24).

4. Pg. 7 Equations 1 and 2: These equations may be better placed in the Methods section.

It is possible to cause misunderstanding regarding our objective if the equations are placed in the Method section because our objective in the study is suggestion of algal growth model to reproduce a temporal change in algal cell abundance in Greenland glacier. Therefore, we described the explanation of the equations in Discussion section.

5. Pg. 8 line 2-3: are these numbers correct? The text states the initial concentration was substantially smaller than the final concentration. Check the concentration numbers.

We checked the concentration numbers in the sentence, but there was no contradiction in the concentration numbers in the sentence. We have revised the sentence because it seems that the previous sentence causes a misunderstanding (Pg 8 lines from 18 to 21).

6. Pg. 8 line 4-5: Why aren't the authors using two separate carrying capacities for Site-A and Site-B, if they have different maximum concentrations of algal cells?

Results suggest that algal cell concentration at Site-A reached to carrying capacity, but it at Site-B continued to increase significantly. The continuous increase of algal abundance suggests that the carrying capacity did not limit affect the algal growth at Site-B. The algal cell concentration at Site-B is likely to increase gradually after day 215 because the calculated snow surface temperature at Site-B was above 0°C after the day. The maximum concentration of algal cell at Site-B possibly close to the carrying capacity at Site-A after day 215. For this reason, we assumed that the carrying capacity at Site-B is a same value of it at Site-A in this study although the carrying capacity may vary among sites. We have added the explanation about the carrying capacity at Site-B to the manuscript (from Pg 8 line 30 to Pg 9 line 2).

7. Pg. 8 line 21-22: The text of 100 times more at Site-A than Site-B is redundant to the previous few lines of text.

We have revised the sentence (Pg 9 lines from 17 to 19).

8. Pg. 24 Fig. 7b and c: Error bounds are needed for the logistic model (solid) line. Similarly, for Fig. 8b and c.

The variance of the algal cell concentration calculated by the logistic model probably increase over time. Since the confidence interval (error bound) possibly be affected by the variance, we consider it will be difficult to calculate the confidence interval. From this reason, we did not estimate the confidence interval for the logistic model line in the study. We have added the explanation about the confidence interval to the manuscript (Pg 9 lines from 5 to 6).

---

## Author Comment (AC3) · 27 Mar 2018

[revised manuscript text omitted]
. Snow pit samples showed that any algal cell was not detected in all of the snow layers before the first appearance of snow algae on the surface. Therefore, the algal cells are unlikely to have originated from beneath the snow, but from the atmosphere. Alternatively, algal cells might have been transported from the ground surface surrounding the glacier or from distant sources. Previous studies reported that mineral

10   dust on glaciers in northwest and southwest Greenland is likely to be supplied from local ground surfaces (e.g. moraine near the glacier), rather than the distant areas (Nagatsuka et al., 2014; 2016). The algal cell spores may be on the moraine near the glacier because the algal cells are probably washed out of the glacier (e.g. moraine) when snowpack disappeared. Therefore, the source of the algal cell spores may be the ground surface near the glacier.

Meteorological records suggest that the initiation of algal growth requires the air temperature to remain above 0°C for a

15   certain period of time after the previous snowfall. The snow algae at both Sites-A and B appeared two days apart from each other. Prior to algal appearances, the hourly air temperature remained above 0°C for 94 h from day 175 at Site-A and for 136 h from day 176 at Site-B; there was no snowfall during this time at either site. The period from the last snowfall appears to be important in initiating snow algal growth, as fresh snow coverage inhibits photosynthesis of the snow algae under the snow. Additionally, snowmelt is required for the initiation of algal growth (Fukushima, 1963; Onuma et al., 2016). Snow

20   algae on a snowpack in Japan has been reported to appear when air temperatures exceed 0°C for 24 h, which is likely the minimum requirement for initiating snow algal growth (Onuma et al., 2016). The duration was longer in this study than that which was observed in Japan. The longer duration may be due to a difference of algal species or weather conditions on this glacier. These results suggest that continuous melting for a minimum of 94 h is required for the initiation of algal growth on the Qaanaaq Glacier although further studies are necessary to determine the snow physical conditions for the initiation.

**4.2 Approximation of the algal growth curve with a numerical model**

In order to reproduce the observed algal growth with a numerical equation, we applied a logistic model that utilizes a general differential equation of microbial growth to the observed algal growth curve. An increase in microbial cells can simply be expressed by a differential equation known as the Malthusian model, which is defined by an initial cell concentration and

30   algal growth rate (Lavoie et al., 2005). The Malthusian model is based on the assumptions that microbial abundance increases by cell division of all present cells at a constant rate, that there is no addition or removal of cells in the habitat, and that light, nutrients, and habitable space are unlimited. According to this model, the microbial growth curve is calculated as follows (Cui and Lawson, 1982):

$$X = X_0 e^{\mu(t-t_0)},\tag{1}$$

where $X$ and $X_0$ are population densities of microbes at t and $t_0$, respectively, and $\mu$ is the growth rate of microbes in $t^{-1}$. The Malthusian model has been applied to observational microbial abundances in sea ice (Lavoie et al., 2005) and in snowfield (Onuma et al., 2016). However, the algal abundance at Site-A did not significantly increase after late July, despite the air temperature remaining above 0°C and a lack of snowfall, indicating that the Malthusian model could not represent the algal growth curve on the surface snow of the Qaanaaq Glacier. The decreased growth rate observed on the glacier suggests that algal abundance has a limited capacity in this habitat. A logistic model is a microbial growth equation with a carrying capacity, and thus could represent the algal growth curve observed in this study. The temporal change of the logistic model is represented as follows (Cui and Lawson, 1982):

$$X = \frac{K}{1+\frac{K-X_0}{X_0}e^{\mu(t_0-t)}}, t = d - d_f,\tag{2}$$

where $K$ is the carrying capacity of algae in the snow surface (depth = 2 cm) and $t_0$ is the day of the first appearance of algae on the snow surface. Since snow algae can grow only on the melting snow surface, we assumed that algal growth was interrupted when snow surface temperature was below 0°C. Thus, $t$ represents the number of the days during which the mean temperature was above 0°C. This equation was fitted to the observational algal cell concentrations at Sites-A and B through Poisson regression. The observational data used are from the day of algal appearance (days 179 at Site-A and 181 at Site-B, $t_0$) through the last day of the study period (days 214 at Site-A and 215 at Site-B, $t_{max}$). This regression is based on the assumption that there is no inflow or outflow of algal cells on the snow surface. The algal abundance on surface snow is possibly affected by algal cells supplied by wind. However, the effect by wind-delivery of algal cells on algal abundance appeared to be small enough to calculate the algal growth because the initial concentration of algae on the surface ($3.1 \times 10^3$ cells m$^{-2}$ on day 179 at Site-A and 7.4 cells m$^{-2}$ on day 181 at Site-B) was substantially smaller than the final concentration ($3.5 \times 10^7$ cells m$^{-2}$ on day 214 at Site-A and $5.0 \times 10^5$ cells m$^{-2}$ on day 215 at Site-B). Although the algal abundance on surface snow is possibly affected by algal cells supplied by cells appeared from subsurface snow as melting proceeded, the effect is unlikely due to the result that any algal cells were not detected in all of the snow layers before the first appearance of snow algae on the surface. The algal cell concentrations in the study sites gradually increased with snow melting except for the period from day 201 to 214 at Site-A, suggesting that the outflow of algal cells by melt water is unlikely to affect the algal cell abundance on the snow surface. To fit Poisson regression to the observed algal cell concentrations, carrying capacity was assumed to be $3.5 \times 10^7$ cells m$^{-2}$ at both sites based on the observed maximum concentration of algal cells (day 214 at Site-A). Although it is uncertain whether the algal concentration on day 214 at Site-A was the greatest during the summer, the carrying capacity on the glacier was likely around $3.5 \times 10^7$ cells m$^{-2}$ since the cell concentrations of all of algal types hardly increased from day 201 to 214 despite air temperatures remaining above 0°C. The algal cell concentration at Site-B continued to increase significantly until day 215, suggesting that the carrying capacity did not limit the algal growth at the site. The algal cell concentration at Site-B would increase further after day 215 because the calculated snow surface temperature at Site-B was above 0°C after the day. The maximum concentration of algal cell at Site-B possibly close to the

carrying capacity at Site-A after day 215. Although the carrying capacity may vary among sites, the carrying capacity at Site-B was assumed to be a same value at Site-A in this study.

Fitting the data to the model showed that the coefficients of determination of the regression ($R^2$) were 0.64 and 0.96 at Sites-A and B, respectively, suggesting that the algal growth curve was reproduced with the equation (Table. 1, Figs. 7 and 8). The confidence intervals for the algal growth curve were not calculated because the confidence intervals possibly be affected by the variance of the algal cell concentration increased over time. The calculated growth curve at Site-A did not reproduce the reduction of algal cell concentration from days 201 to 208. This is likely the reason for the lower $R^2$ value at Site-A. However, the calculated algal cell concentration ($3.4 \times 10^7$ cells m$^{-2}$) was consistent with the observed abundance ($3.5 \times 10^7$ cells m$^{-2}$) at Site-A on day 214, which was the day when algal cell concentration on surface snow was the greatest during the observational period; this suggests that the model can accurately reproduce the timing when algal cell concentration reached the carrying capacity.

**4.3 Factors affecting parameters of algal growth model**

[revised manuscript text omitted]

various glaciers and snowfields. The validation and calibration of the model parameters is important to apply the logistic model to other regions. Recently study reported that algal cell abundance in snowfield of Alaska was estimated by satellite images (Ganey et al., 2017). Such a method may be able to validate and calibrate the algal growth model in various regions. More research is necessary to understand the process of algal growth because other factors, such as nutrient concentration of snow, possibly affect snow algal growth. However, our results demonstrate that a simple numerical model could simulate the temporal variation in algal abundance on snow surface in Greenland. In future, a regional snow physical model (Niwano et al., 2018) coupled with the algal growth model may enable to estimate snow melting regarding the effect of algal blooming and the algal growth model may supply useful information, such as a timing of algal blooming, for the study on the life cycle of snow algae.

[revised manuscript text omitted]

---

## Author Response (AR1)

April 24, 2018

Dr. Yukihiko Onuma

Institute of Industrial Science, University of Tokyo

Chiba, 277-8574, Japan

Phone: +81-4-7136-6965

E-mail: onuma@iis.u-tokyo.ac.jp

Prof. Tedesco

Editor, The Cryosphere

Dear Prof. Tedesco,

We would appreciate the time and effort you have dedicated to providing insightful feedback on ways to strengthen our paper. Please see enclosed our responses to the all reviewers' comments as well as the revised marked-up manuscript entitled as "Observations and modelling of algal growth on a snowpack in northwest Greenland" by Yukihiko Onuma et al. [Paper # tc-2017-252] submitted to the journal The Cryosphere. We have revised the manuscript according to the all reviewers' comments. We hope that the revised manuscript is suitable for publication. We hope for your favorable reply.

Sincerely yours,

Yikihiko Onuma and co-authors

**Reply to Reviewer#1**

The submitted manuscript aims to address a worthwhile gap in our knowledge of snow algae – namely the ability to model the growth of algal blooms over a season. The paper combines empirical observations with a simple growth model, building upon previous work by the same group that applied a Malthusian model to explain the population dynamics of the snow algal bloom by adding a carrying capacity. I enjoyed reading the paper and appreciated the careful field measurements; however, I have two queries for the authors, along with some minor typographical corrections.

We would appreciate very much a number of constructive comments. We also appreciate that you spent your precious time for us. We glad in being able to hear that you enjoyed reading our paper. Our responses (blue text) to each the reviewer's comment (in black text) were described as follows. We also indicate revisions in the updated manuscript with yellow marker as suggested.

Major Comments:

1.  The model assumes no input or removal of cells, indicating that the population dynamics are dominated by in situ cell proliferation. However, no mention is made of the effects of scavenging by melting snow. The algae are likely to become concentrated onto the snow surface as the snowpack melts, and if this is the case then the authors would have measured an increase in surface biomass simply due to the concentrating effects of snow melt. Can the authors show that this was not the case? Similarly, it is hard to envisage zero cell export from the system. I suspect the snowpack algal population dynamics to be the result of in situ growth, scavenging by snow melt, export in meltwater and a small contribution by wind-delivery. Can the authors provide any data or observations to support all these factors being negligible apart from in situ growth?

As the reviewer pointed out, algal cell abundance is possibly increased by the effects of scavenging or wind-delivery and reduced by the effect of melt water. We conducted additional analysis to discuss the effect of scavenging on algal growth in snowpacks (please see the Figures S1 and S2 in this response letter). The results show vertical profiles in algal cell concentration, snow density and snow temperature in snowpit of the study sites before the first appearance of snow algae on the surface. The vertical profiles showed that all of the snow layers in the study sites did not include any algal cell, indicating that a scavenging by snow melt does not affect a temporal change in algal abundance. The effects of wind-delivery appeared to be small enough to calculate algal growth because the initial concentration of algae on the surface, which was likely due to wind in the study sites, was substantially smaller than the final concentration (Site-A: $3.1 \times 10^3$ cells m$^{-2}$ vs. $3.5 \times 10^7$ cells m$^{-2}$, Site-B: 7.4 cells m$^{-2}$ vs. $5.0 \times 10^5$ cells m$^{-2}$). The movement of algal cells by

melt water also appeared to be small enough to calculate algal growth because algal cell concentrations in the study sites gradually increased with snow melting except for the period from day 201 to 214 at Site-A. Therefore, we simulated the temporal changes in algal cell concentration using logistic model based on the assumption that there is no inflow or outflow of algal cells on the snow surface. We have revised the manuscript to discuss the effects of scavenging, wind-delivery and melt water on a temporal variation in algal abundance (from pg 8 line 30 to pg 9 line 4). And, we have added the method and result of the snowpit observation to the discussion about the origination of algal cell (pg 4 lines from 4 to 6, pg 6 lines from 13 to 14 and pg 7 lines from 5 to 6).

2. I think the authors overstate the utility of the model. As it stands the manuscript shows that a logistic modelling approach is appropriate for predicting the population dynamics of snow algae for specific field sites; however, insufficient information is provided to enable the application of the model elsewhere. The fact that between these two sites the coefficient of determination for the model applied to observations varies between 0.64 and 0.96 suggests to me that additional site specific factors influence the growth rate. K is, as the authors discuss, dependent upon the nutrient availability and available space, but no quantitative link is drawn between either variable and the value of K. The slope of the model is assumed to be entirely dependent upon time, but it sees unlikely that this assumption would often be satisfied. Presumably, the rate of growth is in reality affected by many more variables. For example the study by Onuma et al. (2016) found algal blooms to initiate just 24 hours after melting began which the authors attribute to either a different algal species (can you confirm this with observations?) or 'weather conditions'. This suggests that the growth rate varies between sites and cannot be assumed constant. As I understand it, weather conditions are precisely what you are trying to analyse as potential drivers of algal growth in this paper, so it seems contradictory to invoke them as an additional source of uncertainty. It seems likely that the initiation and growth rate of snow algal blooms are the result of a combination of meteorological factors plus site specific variations such as nutrient availability in the snowpack that may well be difficult to unpick, limiting the predictive capability of the model on other snowpacks. Can the authors provide any more data or discussion to support the applicability of the model?

We also consider that there probably is an insufficient information for applying of logistic model to various fields. The model requires three parameters, which are initial cell concentration, growth rate and carrying capacity, to calculate temporal change in algal cell abundance, and they are likely to vary in different snow fields or regions. The initial cell concentration and carrying capacity are likely to be associated with abundance of mineral particle on the snow surface. The carrying capacities in various fields may be determined by approximation using the relationship between observational abundance of algal cells and mineral dust on the snow surface. The growth rate may be similar in other sites as far as the algal species is same, i.e. *Cd.nivalis*. However, further observations are necessary for estimation of the

model parameters in various fields. This study is the first attempt to establish the algal model based on only a single glacier and season, thus, we could not say the feasibility of the model to be applied in more extensive areas. We have weakened the statements of the application to other regions and mentioned only possibilities to extend the glacier and ice sheet (from pg 11 lines from 7 to 14). Also, we have added the explanation about the determination of the carrying capacity in various fields by the approximation to the manuscript as suggested (pg 10 lines from 25 to 26).

Specific Comments:

1. pg 1 line 23: change 'bloom' to 'blooms', change 'changes' to 'change'

The words have been corrected (pg 1 line 23).

2. pg 1 line 24: change 'bloom' to 'blooms'

The word has been corrected (pg 1 line 24).

3. pg 2 line 20: delete 'works'

The word has been deleted (pg 2 line 19).

4. pg 2 line 26: these references are a mixture of ice algae and snow algae. Ice algae has been suggested to enhance ablative losses to the GrIS by reducing albedo (see more recent papers by Tedstone et al. 2017 and Stibal et al. 2017). To my knowledge, evidence for accelerated snow line retreat due to snow algal blooms has not yet been presented for Greenland. I suggest rewording this sentence accordingly or providing specific references for accelerated snow line retreat.

The sentence has been revised as suggested (pg 2 lines from 25 to 27). We have added the explanation about reduction of surface ice albedo due to ice algal bloom. The following reference has been added at pg2 line 27.

Stibal, M., Box, J. E., Cameron, K. A., Langen, P. L., Yallop, M. L., Mottram, R. H., … Ahlstrøm, A. P.: Algae drive enhanced darkening of bare ice on the Greenland ice sheet, Geophysical Research Letters, 44, doi: 10.1002/2017GL075958, 2017.

Tedstone, A. J., Bamber, J. L., Cook, J. M., Williamson, C. J., Fettweis, X., Hodson, A. J., and Tranter, M.: Dark ice dynamics of the south-west Greenland Ice Sheet, The Cryosphere, 11, 2491-2506, doi: 10.5194/tc-11-2491-2017, 2017.

5. Pg2 line 28: The Lutz studies were limited to the visible wavelengths and were not really measuring albedo, but a

proxy. Better to say that they showed algae can modulate snow reflectance in the visible wavelengths.

The sentence has been revised as suggested (pg2 line 28).

6.  Pg3 line 5: superscript km^2

The word has been corrected (pg 3 line 10).

7.  pg4 line 4: normalising to area presumably requires that the algae only inhabit an extremely thin layer on the upper surface of the snow – other studies (e.g. Thomas et al., 1979; Hodson et al., 2017) indicate that subsurface red algae can exist. Are you confident that the algae were confined to the upper surface? Is this supported by your observations?

We do not have algal abundance data in subsurface snow layer after algal appearance in surface snow layer of the glacier. However, the vertical profiles in algal cell concentration in snowpack of the study sites show that there were no algal cells in snowpack before snow algae first appear in surface snow (Figures S1 and S2) as the response to major comment 1. In addition, snow algal cells are likely to be supplied from the atmosphere on surface snow in the study sites. These results suggest that snow algal cells concentrated to surface snow although a part of the algal cells may be flowed to subsurface snow by the melt water. We have revised the manuscript to discuss the effect of algal cells in subsurface snow on algal growth in surface snow (pg 9 lines from 1 to 2).

8.  Pg6 line 21: change 'active radiation' to 'photosynthetically active radiation'

The words have been changed (pg 6 line 31).

9.  pg 7 line 24: delete 'can'

The word has been deleted (pg 8 line 11).

10. pg 8 line 33: statistical significance should be supported by test name and values.

We have added a result of statistical test (Student's $t$-test) to the sentence (pg 10 lines from 2 to 3).

11. pg8 line 28: change 'snowfileds' to 'snowfields'

The word has been corrected (pg 9 line 30).

[Figure]

Figure S1. Vertical profiles of snow algal abundance and physical properties in a snow pit on day 162 at Site-A. (a) algal cell concentration, (b) snow density, (c) snow temperature in snow. Standard deviation shown by error bars.

[Figure]

Figure S2. Vertical profiles of snow algal abundance and physical properties in a snow pit on day 168 at Site-B. (a) algal cell concentration, (b) snow density, (c) snow temperature in snow. Standard deviation shown by error bars.

**Reply to Reviewer#2**

Summary:

The manuscript tries to understand the link between snow algae growth and its relationship with albedo and ablation rates. The authors use field observations to monitor changes in snow algae over the melt season and apply these results to a model to simulate algae blooms. This research is relevant to further understanding of how algae abundance on snowpacks evolution over a season and what effects they have on surface albedo and melt rates. While I believe this work is relevant to the community, I think the manuscript could be written in a more compelling way, with greater connections and applicability to the Greenland ice sheet. And, while the model serves its purpose for this study, I think its functionality should not be overstated. And, the linkage with snow algae to surface albedo and melt rates is not made in the manuscript. I think greater emphasis and connection with albedo and melting should be added. And, what these observation and modeling efforts mean for implementation into regional climate models.

We would appreciate very much a number of constructive comments. We also appreciate that you evaluated our approach to further understanding of temporal change in algal abundance on snowpacks although our manuscript needs more revising. Our responses (blue text) to each the reviewer's comment (in black text) were described as follows. In addition, we indicate revisions in the updated manuscript with yellow marker as suggested.

Major Comments:

2.  How representative is the model for use in other regions, beyond a glacier ice cap? Can the numerical model be feasibly used elsewhere on the ice sheet?

The model requires three parameters, which are initial cell concentration, growth rate and carrying capacity, to calculate temporal change in algal cell abundance, and they are likely to vary in different snow field or regions. We would think that the growth rate may be similar in other sites as far as the algal species is same, i.e. *Cd.nivalis*. The other two parameters, which are the initial cell concentration and carrying capacity, are likely to be associated with abundance of mineral particle or chemical properties on the snow surface, however, further studies are necessary. This study is the first attempt to establish the algal model based on only a single glacier and season, thus, we could not say the feasibility of the model to be applied in more extensive areas. We have weakened the statements of the application to other regions and mentioned only possibilities to extend the glacier and ice sheet (from Pg 11 lines from 7 to 14).

3.  What are the larger impacts of this study? I think the authors should discuss this further and link the field and

modeling study to broader application and regions of the Greenland ice sheet.

We established this algal model to quantify the algae on the entire Greenland Ice Sheet and to evaluate their impact on surface albedo and melting rate of the snow. We plan to couple this model with a regional snow physical model (e.g. Niwano et al., 2018) to simulate snow albedo in future. Also, the model would be useful to know the algal life cycle on the ice sheet. We have added more detailed explanation on the broader applications of the model in the text (Pg 11 lines from 14 to 17) with the following reference.

Niwano, M., Aoki, T., Hashimoto, A., Matoba, S., Yamaguchi, S., Tanikawa, T., Fujita, K., Tsushima, A., Iizuka, Y., Shimada, R. and Hori, M.: NHM–SMAP: spatially and temporally high-resolution nonhydrostatic atmospheric model coupled with detailed snow process model for Greenland Ice Sheet, The Cryosphere, 12, 635–655, https://doi.org/10.5194/tc-12-635-2018, 2018.

4. There appears to be large uncertainty associated with the algae cell observations (Fig. 7b). How can the authors argue that a good fit is achieved between the field and modeled algal cell concentration? There needs to be further discussion on the utility of the logistic model as well as its deficiencies. How can we improve the model? What data and additional variables are needed? And, what is the greater link to surface albedo and melting?

We agree the model has a large uncertainly, particularly in the late of the melting season. Although our model has an uncertainly, it simulated the timing of algal blooming and the other of magnitude of algal abundance well. Again, this study is the first attempt to establish the algal model based on only a single glacier and season. More data of temporal change of algal abundance could reduce the uncertainty. Other factors affecting algal growth (e.g. inflow or outflow of algal cells in snowpack) may improve the model, however, we would not propose more complex model because of utility for coupling with the climate model. We'll try to simulate a temporal change in snow albedo coupling the algal model with a snow physical model (e.g. Aoki et al., 2011; Niwano et al., 2012) in the future. We have added the discussion of uncertainty and application of the model in the manuscript to reflect reviewer's comments (from Pg 2 lines from 31 to 34, Pg 9 line 14 and Pg 11 lines from 7 to 14). The following references have been added at Pg 2 line 32.

Aoki, T., Kuchiki, K., Niwano, M., Kodama, Y., Hosaka, M., and Tanaka, T.: Physically based snow albedo model for calculating broadband albedos and the solar heating profile in snowpack for general circulation models, J. Geophys. Res., 116, D11114, https://doi.org/10.1029/2010JD015507, 2011.

Niwano, M., Aoki, T., Kuchiki, K., Hosaka, M., and Kodama, Y.: Snow Metamorphism and Albedo Process (SMAP) model for climate studies: Model validation using meteorological and snow impurity data measured at Sapporo, Japan, J. Geophys. Res., 117, F03008, https://doi.org/10.1029/2011JF002239, 2012.

Specific Comments:

12. Pg. 6 line 2: Change to '3.1*10^3 cells m^-2'. And, again on line 4.

The words have been corrected (Pg 6 lines 11 and 14).

13. Pg. 6 line 17-18: What evidence do you have to validate that the red algal cells originate from windblown spores? Is there a way to verify this further and possible local sources (eg. nearby tundra)?

We don't have a direct evidence to prove the source of algal spores. A study on mineral dust on this glacier showed that they are likely to be supplied from local ground surface (e.g. moraine near the glacier), rather than the distant areas (Nagatsuka et al., 2014; 2016). As the initial algal concentration was greater in the site where the mineral dust was more abundant, we would think that the algal spores may be originated from the same as mineral dust. We have discussed the source of the algal spores more carefully in the text (Pg 7 lines from 8 to 11) and the following references have been added at Pg 7 line 9.

Nagatsuka, N., Takeuchi, N., Uetake, J. and Shimada, R.: Mineralogical composition of cryoconite on glaciers in northwest Greenland. Bull. Glaciol. Res., 32, 107–114, doi:10.5331/bgr.32.107, 2014.

Nagatsuka, N., Takeuchi, N., Uetake, J., Shimada, R., Onuma, Y., Tanaka, S. and Nakano, T.: Variations in Sr and Nd isotopic ratios of mineral particles in cryoconite in western Greenland. Front. Earth Sci., 4, 93, doi: 10.3389/feart. 2016.00093, 2016.

14. Pg. 7 line 5-6: reword sentence structure.

The sentence has been revised (Pg 7 lines from 22 to 24).

15. Pg. 7 Equations 1 and 2: These equations may be better placed in the Methods section.

We thank this suggestion and agree that is another option to present our study. However, our study has not only the establishment of the model, but also describe the observational results of algal temporal change on the glacier. So, we finally decided to keep the equations to be placed in discussion section, as we would think it would be readable.

16. Pg. 8 line 2-3: are these numbers correct? The text states the initial concentration was substantially smaller than the final concentration. Check the concentration numbers.

We checked the concentration numbers in the sentence, but there was no contradiction in the concentration numbers in the sentence. We have revised the sentence because it seems to cause a misunderstanding (from Pg 8 line 31 to Pg 9 line 1).

17. Pg. 8 line 4-5: Why aren't the authors using two separate carrying capacities for Site-A and Site-B, if they have different maximum concentrations of algal cells?

Results showed that the increasing rate of the algal concentration at Site-A appeared to be slow down at the end of the study period, in contrast, that at Site-B continued to increase significantly. Thus, we considered that algal concentration did not reach the carrying capacity at Site-B. The algal cell concentration at Site-B would increase further after the study period because the calculated snow surface temperature at Site-B was above 0°C after the day. Therefore, we used the only carrying capacity at Site-A. We have added the explanation about the carrying capacity at Site-B to the manuscript (Pg 8 lines from 25 to 29).

18. Pg. 8 line 21-22: The text of 100 times more at Site-A than Site-B is redundant to the previous few lines of text.

We have revised the sentence (Pg 9 lines from 23 to 25).

19. Pg. 24 Fig. 7b and c: Error bounds are needed for the logistic model (solid) line. Similarly, for Fig. 8b and c.

We may add the uncertainly of the model as error bounds for the line, however, it is difficult to quantify the uncertainly (confidence level) for the logistic model line in the study. The variance of the algal cell concentration calculated by the logistic model probably would not be constant, but increase over time, therefore, the confidence level (error bound) is likely to vary and is too complicate to be reproduce quantitatively with the model. We have added the explanation about the confidence level to the manuscript (Pg 9 lines from 7 to 9).

[revised manuscript text omitted]
. The calculated growth curve at Site A did not reproduce the reduction of algal cell concentration from days 201 to 208. This is likely the reason for the lower R$^2$ value at Site-A. However, the calculated algal cell concentration ($3.4 \times 10^7$ cells m$^{-2}$) was consistent with the observed abundance ($3.5 \times 10^7$ cells m$^{-2}$) at Site A on day 214, which was the day when algal cell concentration on surface snow was the greatest during the observational period; this suggests that the model can accurately reproduce the cell abundance in the order of magnitude and the timing when

15   algal cell concentration reached the carrying capacity.

**4.3 Factors affecting parameters of algal growth model**

[revised manuscript text omitted]